# Beyond Labeling Oracles - What does it mean to steal ML models?

**Avital Shafran**[*]                                                      *avital.shafran@mail.huji.ac.il*
*The Hebrew University of Jerusaelm*

**Ilia Shumailov**[†]                                                      *ilia.shumailov@chch.ox.ac.uk*
*University of Oxford*

**Murat A. Erdogdu**                                                      *erdogdu@cs.toronto.edu*
*University of Toronto & Vector Institute*

**Nicolas Papernot**                                                      *nicolas.papernot@utoronto.ca*
*University of Toronto & Vector Institute*

**Reviewed on OpenReview:** *https://openreview.net/forum?id=950naKZIyh*

## Abstract

Model extraction attacks are designed to steal trained models with only query access, as is often provided through APIs that ML-as-a-Service providers offer. Machine Learning (ML) models are expensive to train, in part because data is hard to obtain, and a primary incentive for model extraction is to acquire a model while incurring less cost than training from scratch. Literature on model extraction commonly claims or presumes that the attacker is able to save on both data acquisition and labeling costs. We thoroughly evaluate this assumption and find that the attacker often does not. This is because current attacks implicitly rely on the adversary being able to sample from the victim model's data distribution. We thoroughly research factors influencing the success of model extraction. We discover that prior knowledge of the attacker, *i.e.* access to in-distribution data, dominates other factors like the attack policy the adversary follows to choose which queries to make to the victim model API. Our findings urge the community to redefine the adversarial goals of ME attacks as current evaluation methods misinterpret the ME performance.

## 1 Introduction

Modern ML models are valuable intellectual property, in part because they are expensive to train (Sharir et al., 2020; Patterson et al., 2021), and model extraction (ME) attacks threaten their confidentiality. In an ME attack, the adversary attempts to steal a victim model over query access, in order to obtain an approximate copy of that model with similar performance (Tramèr et al., 2016). In the most common class of ME attacks, the adversary uses queries to the victim model as a training set for obtaining a copy of that model[1].

ME attacks are a looming threat for ML-as-a-Service providers, whose business model depends on paid query access to their models over APIs. In fact, Kumar et al. (2020) outlines that ME is one of the most concerning threats to the industry. Typically, the claimed motivation to conduct ME attacks is to avoid the costs involved in training a model from scratch. Most notably, this refers to the *data collection* and *data labeling* costs, and sometimes mentions the *training* computational costs. The implicit assumption

---

[*]Work performed while the author was visiting the Vector Institute.
[†]Work done for the most part while the author was a Postdoctoral Fellow at the Vector Institute.
[1]Figure 6 illustrates such a typical attack, while other types of ME attacks are discussed in Section 2.

is that ME is data- and/or compute- cheaper (Jagielski et al., 2020) – the adversary's goal is to get the best performing model using the least resources. This includes using easily available queries, possibly from another distribution than the victim's as well as minimizing the query complexity, *i.e.* stealing the model using less samples than was used to train it in the first place. Intuitively, this query efficiency criterion can be possible, provided the attacker identifies the most informative queries from the available query distribution.

In this paper we investigate primary assumptions and arguments made in the ME literature. We find that current research does not adequately account for how the adversary's prior knowledge contributes to their ability to choose the right queries to make. Many existing attacks demonstrating impressive performance assume the attacker has access to either a small number of unlabeled samples from the true victim's training distribution (thereafter, in-distribution data or IND), or a potentially larger number of unlabeled samples from a different, yet similar, distribution (thereafter, out-of-distribution data or OOD) (Correia-Silva et al., 2018; Orekondy et al., 2019a; Dziedzic et al., 2022; Jagielski et al., 2020; Pal et al., 2020; Zhang et al., 2021; Okada et al., 2020; Karmakar & Basu, 2023). In contrast, attacks that do not assume any prior knowledge, and perform in a *data-prior free* manner, usually exhibit a drastically higher query complexity (Truong et al., 2021; Jagielski et al., 2020).

We extensively experiment with ME attacks using additional control mechanisms to disambiguate the effect of prior knowledge from the attack policy. We find that performance of current ME attacks is dominated by the IND data; in settings where IND is mixed with OOD data to decrease the data costs, the impact of OOD is minor; moreover, when only IND is used, we find that the victim model merely serves as a labeling oracle, and not necessarily the most cost-effective one. **Overall, our position is that the practical threat posed by ME attacks is often exaggerated since the attacker bears comparable costs to the victim.**

To summarize, we make the following contributions:

- We demonstrate how prior knowledge dominates ME performance and cost efficiency, even in cases where OOD data is used to augment the attacker's query distribution, thus leaking information about IND decision boundaries at lower acquisition costs.

- By modifying victim models such that the OOD leakage is reduced, we further demonstrate the dependence on prior knowledge, which can only be compensated by increasing the query complexity.

- We show that if the adversary has access to even just a small amount of IND data, ME becomes a labeling oracle. In other words, aside from providing a label, victim labels leak limited information about the decision boundaries.

## 2 Related work

Model extraction attacks, also called model stealing, were first explored by Tramèr et al., who proposed an attacker that queries the victim model to label its dataset and trains a model to match these predictions. Various works extended the attack with the goal of achieving similar *task accuracy* while minimizing the number of queries required. Most of these attacks either assume access to a surrogate dataset (Correia-Silva et al., 2018; Orekondy et al., 2019a) or to a portion of the real training set (Rakin et al., 2021), or use random queries (Krishna et al., 2019; Chandrasekaran et al., 2020) in domains like NLP. We discuss the relation between ME to active learning in Appendix A. Truong et al. attested that, in order to successfully extract the victim model, the attacker's dataset must share semantic or distributional similarity to the real training dataset; otherwise resulting in an insufficient extraction accuracy. To mitigate this issue, they propose a data-free model extraction attack (DFME) that does not require a surrogate dataset. Inspired by data-free knowledge distillation (Lopes et al., 2017), they train a generative model to synthesize queries which maximize disagreement between the attacker's and the victim's models. A similar method was also proposed by MAZE (Kariyappa et al., 2021). (Lin et al., 2023) investigated ways to reduce the query complexity of data-free approaches. These results align with our analysis and show that the adversary can compensate for the absence of prior knowledge by using a very large query budget.

Jagielski et al. proposed an ME attack that, in addition to *accuracy*, targets *fidelity*. In this scenario, the attacker focuses on the exact reproduction of the victim model behaviour on all possible inputs up to symmetries. Carlini et al. argued that ME attacks are essentially a cryptanalytic problem, and proposed an attack based on differential cryptanalysis, with the goal of high fidelity extraction. Note that this line of research falls outside the scope of our work, since both the worst and the average-case query complexities reported by both papers significantly exceed accuracy-based ME attacks empirically. As such, in this work we will focus our evaluations on ME attacks that target task accuracy.

While the most commonly used threat model for ME attacks assumes black-box query access to the victim model, it is important to note that there is a significant line of work which focuses on side-channel ME attacks (Zhu et al., 2021; Hu et al., 2019; Yan et al., 2020; Hua et al., 2018; Xiang et al., 2020; O'Brien Weiss et al., 2023; Duddu et al., 2018). This line of work is out of scope for our paper since most side-channels are fixable with careful system re-design, while ME attacks should remain unaffected as long as model provides genuine responses to queries.

It is worth mentioning that in the limit there is little that a victim can do to stop ME. Unlimited data sampling allows for exhaustive search of all possible inputs, whereas functions that are used to approximate decision boundaries in the limit can approximate arbitrary functions (Hornik et al., 1989). This leads to an inherent performance trade-off, where the victim sacrifices performance of the model to limit information leakage to an acceptable level.

## 3 Background

In this section, we first discuss the different costs involved in model extraction and connect them with the ME literature; we cover related works in detail in Section 2. We then describe how prior literature approached cost reduction using random data, why this can work theoretically, and what assumptions are required. Subsequently, we describe the additional mechanism we use to disambiguate the effect of the attack policy from increased query budget in ME.

### 3.1 Definitions

In what follows, we refer to in-distribution data as IND and out-of-distribution data as OOD. Note that we follow the conventions of the field and use these terms loosely to describe the distribution that training data comes from for the victim model as IND, while OOD refers to any data that comes from a different distribution.

We define an ME adversary as 3-tuple $(\mathcal{D}_{\mathsf{IND}}, \mathcal{D}_{\mathsf{OOD}}, \Pi)$, where $\mathcal{D}_{\mathsf{IND}}$ represents the adversary's access to the IND, *i.e.* it's prior knowledge, $\mathcal{D}_{\mathsf{OOD}}$ represents the adversary's unrelated OOD query distribution, and $\Pi$ represents the adversary's query selection policy. When attacking, the adversary uses $\Pi$ to select samples to be queried from its IND and OOD distributions. Then, the queries are issued to the victim model to obtain labels. These labels can be either full probability vectors, top-k probabilities, or label-only. In our evaluations, we focus on the stronger setting of full probability vectors. At last, the adversary trains a model on the labeled query set. Importantly, since attacks in the current literature vary all three from the above, it is hard to pinpoint why a given attack seemingly performs better – could it be a better attack policy or perhaps more informative data? This is the question we answer.

### 3.2 Primer on ML costing

We are not aware of any ME literature that formally defines what makes model extraction attacks successful. While high-fidelity extraction attackers explicitly expect to extract a model within a given error, intuitively, accuracy-driven ME can be considered successful if stealing a model costs less than developing it. That is, as long as the accuracy of the stolen model matches the victim it is a success. The overall cost of producing and deploying a model can be broken down into three main parts: (1) data collection, (2) data labeling, and (3) model training. We note that there are other costs corresponding to ML infrastructure, however they are not relevant to the current ME threat model.

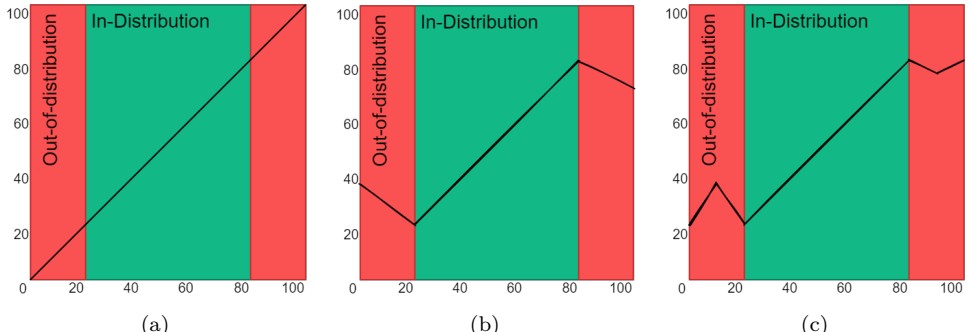

Figure 1: Consider a linear classifier for which the decision boundary is given by the line $y = \alpha x$. An attacker attempts to steal the model (*i.e.* find the corresponding $\alpha = 1$ from the example above). The green region $x \sim [20, 80]$ is in-distribution behaviour that the attacker wants to replicate, the red region $x \sim [0, 20] \cup [80, 100]$ is out of distribution and is not important for the task. The left case requires a single parameter to be approximated, the middle needs 5, whereas the right requires 9.

Now we turn to CIFAR10 as a cost case study – note that CIFAR10 is one of the most evaluated benchmarks for ME. Here we assume CIFAR10 dataset with 60k samples that can be labeled for $35\$ * 50 + 10 * 25\$ = 2000\$$ with Google Cloud (Cloud, 2024) and $0.04\$ * 60000 = 2400\$$ with Amazon Sagemaker (Amazon, 2024). Data collection here is practically free, since per category one can scrape the internet freely. We model the defender cost as $c = n * (pl + cc)$ and the attacker cost as $c_a = n_a * (pl_a + cc_a)$, where $n$ is the number of points to annotate, $pl$ is the per-label cost and $cc$ is the data collection cost. We assume that the attacker wants to succeed in the attack and thus attacker's cost is less than that of a defender $n * (pl + cc) > n_a * (pl_a + cc_a)$. First, we consider the attacker who extracts a model with DFME with 20 million queries. If we assume that the defender used Google Cloud to label their data, we find that the attacker's labels have to cost less than $0.00012\$$:

$$20M * pl_a < 2,400\$ \rightarrow pl_a < 0.00012\$ .$$

The queries in DFME are synthetic, therefore there is no data collection cost ($cc_a = 0$). Next, we consider an ME attack that only utilizes $5\% = 3000$ datapoints of prior knowledge, and no additional queries of any sort. Here, attacker breaks even at $0.8\$$ per-label. Note that with 5% of data, as we show later in Section 5.1, it is indeed possible to extract a model, while remaining well within a reasonable budget in the current labeling market. At the same time, extraction with DFME stops being cost-effective when the attacker queries are even minimally priced by the defender.

This case study suggests that the claim over the cost-effectiveness of ME is not so trivial, and should be carefully considered in different use-cases. An attacker must estimate the cost elements defined above, namely sample complexity, data collection, and data labeling, in order to properly justify performing the attack. We now turn to the investigating the cost effectiveness of current ME attacks in the literature.

## 4 Methodology

A common threat model assumes IND data is scarce and expensive to obtain, thus many attackers attempt to reduce their data collection costs through OOD queries. Note that this is the most common way for ME literature to reduce costs. But how might this work? Hyperbolizing, this implies that by asking questions about *e.g.* a 'dog', one may learn something about unrelated concept of a 'building'. In this section, we first provide intuition for when such extraction is possible. We then build on this intuition and design an explicit mechanism to measure how much OOD queries can possibly contribute to attack performance.

**Warm-up: Linear classification** Consider a linear classifier that splits the input space into two decision regions, and the corresponding decision boundary is defined by the line $y = \alpha x$. Here, $(x, y)$ denotes the

two-dimensional input, and $\alpha$ is the single model parameter that determines the entire decision boundary, which we demonstrate in Figure 1. ME in this context is equivalent to estimating the decision boundary over the green IND region where $x \sim [20, 80]$. The attacker is assumed to have the ability to query the model with arbitrary data but has no knowledge of where the IND region starts or ends. We assume that the depicted red region, $x \sim [0, 20] \cup [80, 100]$, is OOD. Note that it is only important to perform well on the task in the IND region, and neither the victim nor the attacker are concerned with making predictions for OOD inputs.

The left-most plot represents the case of a naive victim model with the original decision boundary. Here ME reduces to estimating a single parameter $\alpha$, which can be achieved by using just 2 data points. Note that, in this case, the IND model can be extracted equally well using both IND or OOD data points, as the model behaviour is shared between both domains. In the middle case, we slightly increase the complexity of the decision boundary in the OOD area by introducing a new linear boundary with non-zero intercept in each region. Here, the OOD queries reveal less information over the IND behaviour, thus requiring the adversary to sample from both regions in order to faithfully learn the boundary. As the attacker does not know where the IND and OOD regions are, it must approximate a piece-wise linear boundary, *i.e.* the slopes and the intercepts of the new lines, in addition to the slope of the original IND boundary. This increases the required number of parameters to 5 and the minimal required sample size to 6, and consequently, the cost of ME. In the right-most case, we further increase the complexity by introducing two additional linear decision boundaries per region, resulting in a total of 9 parameters to be estimated using at least 10 data points.

The above example demonstrates that while the IND area was not modified, adding complex and task-independent boundaries to OOD regions can significantly increase the attacker's required capacity and sample complexity, thus making ME more costly. This behaviour is related to the locally independent nature of large models when performing regression or classification tasks. As sampling the boundary at one point does not necessarily reveal any information on the boundary IND, the attacker must explore the entire input domain. The main assumption here is that *the attacker cannot know which areas are important.* In other words, we find that ME attacks implicitly assume that the adversary has prior knowledge of the distribution to be able to reproduce the victim model's predictions on the task of interest. Without such knowledge, the adversary is unable to tell whether making a query to the victim model will aid its goal of learning the true decision boundaries. Hence, we demonstrate that ME adversaries either require prior knowledge about the underlying distribution or need to submit a large number of queries to the victim model.

## 4.1 Sampling complexity intuition

When useful responses are limited to the IND region only, the attacker's success is dominated by the percentage of queries sampled from IND, which is a function of the attacker's prior knowledge of the victim's true distribution. In this scenario, the weakest attacker, with no prior knowledge, is the random guess attacker that samples useful information with probability $|\mathcal{X}_{\text{useful}}|/|\mathcal{X}|$, where $\mathcal{X}$ is the entire input domain and $\mathcal{X}_{\text{useful}}$ is the IND part of the domain. The strongest attacker here has precise knowledge of the IND region within $\mathcal{X}$ and can sample only from this region, which results in samples useful for the attacker's goal. This is the strongest attacker, as no ME defense can truly defend against such an attacker unless it corrupts the utility of the model as it predicts on the IND domain. Instead, the average-case attacker has partial prior knowledge about the true distribution. As such, the probability of sampling IND is a function of this prior, which indicates the overlap between the true distribution and the attackers' query distribution. We estimate the probability of randomly sampling IND for models considered in this section and find it equals 0.00001% for CIFAR10 models and 0.01% for MNLI. We expand on the procedure in Appendix E.

## 4.2 On hardness of OOD detection

Recently, Tramer (2021) demonstrated that robust adversarial example detection is as hard as robust classification. Here, robustness of the detector refers to consistent detection over the $\epsilon$-radius around a data point, representing the maximal distance for an adversarial example. Fundamentally, Tramer demonstrates that robust detection over an $\epsilon$ region around a given (not necessarily adversarial) data point implies, albeit inefficient, an ability to successfully classify for at least $\epsilon/2$ region. One way to reason

about the hardness of sampling from IND is to think about how informative the predictions of the victim model are on OOD queries. Simply put into Bayesian interpretation, Tramer says that an ability to compute $\mathbb{P}_{\mathcal{D}}(x = x_i, y = y_i)$, *i.e.* perform OOD detection robustly with knowledge of the labels, implies an ability to classify robustly $\mathbb{P}_{\mathcal{D}}(y = y_i | x = x_i)$. Intuitively, that follows from the Bayes rule with $\mathbb{P}_{\mathcal{D}}(x = x_i, y = y_i) = \mathbb{P}_{\mathcal{D}}(y = y_i | x = x_i) * \mathbb{P}_{\mathcal{D}}(x = x_i)$, where $\mathbb{P}_{\mathcal{D}}(x = x_i)$ represents prior knowledge over the true data distribution $\mathcal{D}$, *i.e.* the likelihood of $x_i$ coming from $\mathcal{D}$. In this paper we do not make assumptions about the labels – one can certainly imagine an attacker who has access to some labeled data. Do note however that label information is not necessary for a classic OOD detection *i.e.* checking if a given point comes from a given distribution $\mathbb{P}_{\mathcal{D}}(x = x_i)$. There are two possibilities. First, if the attacker knows the labels of the points being queried, then the setting is exactly the same as the one considered by Tramer *i.e.* robust OOD detection is reducible to robust classification. Second, if the attacker is assumed to be capable of estimating $\mathbb{P}_{\mathcal{D}}(x = x_i)$ robustly, reduction to classification without labels is impossible – the attacker would simply need to use the model as a labeling oracle.

Practically, this implies that with the OOD informativeness controlled, ME might not be the most efficient way for the attacker to achieve their objectives, given the capabilities they have access to. If they have the labels and an OOD detector, then they can already perform classification. If they do not have the labels, they can, at most, get the benefit of labeling their dataset, after which they get classification. If they are unable to solve the OOD detection problem, they would have to first build an OOD detector (which means collecting costly datapoints) since otherwise, OOD queries provide them no information about the IND region. In other words, once we take into account the OOD informativeness control, a **model extraction attacker is only as good as their knowledge of the underlying dataset; but if this knowledge includes labels, often there is no need to extract models in the first place.** This allows us to evaluate the performance of the attacker policy relative to the IND.

### 4.3 Out-of-Distribution instrumentation

In Section 5 we empirically evaluate our hypothesis that ME attacks make an implicit assumption about the usefulness of OOD queries. We show that when this assumption does not hold, and the OOD region is not indicative of the IND behaviour, ME attack success is reduced to being a function of prior knowledge. Due to space limitations, we only briefly describe the details of the model instrumentation, and refer the reader to Appendix F for full details.

Given the original victim model $\mathcal{V}_o$, we create a hybrid victim model $\mathcal{V}_h$ by combining $\mathcal{V}_o$ with an additional module $\mathcal{V}_f$ with different, or additional, decision boundaries. This additional model is used to provide predictions for OOD queries that differ from the predictions they would have gotten from the original model. For each query $x$, the hybrid model $\mathcal{V}_h$ applies some decision rule $R$ to classify $x$ as IND or OOD, and uses the corresponding model for prediction. We design the additional model $\mathcal{V}_f$ such that the decision boundaries of both models would have similar smoothness properties; thus, learning the "fake" boundaries is expected to be of the same level of difficulty, with nearly statistically indistinguishable output distributions. We discuss this further in Appendix F.4. For the decision rule $R$, we apply some pre-defined threshold $\tau$ over the prediction confidence of $\mathcal{V}_o$, with a softmax temperature of 2 to better calibrate $R$. If the confidence is higher than $\tau$, the hybrid model returns $\mathcal{V}_o(x)$, otherwise it returns $\mathcal{V}_f(x)$. Prediction confidence serves as a naive OOD detector (Hendrycks & Gimpel, 2016; DeVries & Taylor, 2018) and represents one of the simplest settings for the attacker. More advanced OOD detectors will strictly improve the separation of IND and OOD queries, therefore decreasing the number of false negative queries, *i.e.* OOD queries that are classified IND and are provided with a meaningful prediction by $\mathcal{V}_o$. For the fake model $\mathcal{V}_f$ we fit a Gaussian Mixture Model (GMM) for each class, and split the input domain into centroids defined by an anchor point. These are used to "assign" queries to one of the GMMs for prediction with a random yet consistent class prediction around anchor point.

For a given query sample $x$, we compute its feature representation and find the nearest $L_2$ anchor point. We sample a fake prediction $\tilde{y}$ using the GMM corresponding to the chosen anchor point and return it as the fake model prediction $\mathcal{V}_f(x) = \tilde{y}$. The construction of the GMMs and their anchor points, discussed in detail in Appendix F, results in OOD samples being "wrongly" predicted while still exhibiting similar smoothness

properties. Put altogether, we have presented here a method to control the usefulness of the victim model's predictions on OOD queries – which will allow us to ablate its effect on the performance of ME.

We note that the principles of our instrumentation are similar to those used by the defense proposed by Kariyappa & Qureshi (2020), however the exact construction of $\mathcal{V}_f$ differs, where Kariyappa & Qureshi train a separate model to provide incorrect responses, while we sample statistically plausible ones.

## 5 Evaluation

In the next subsections sections we empirically answer the following five questions:

- (Section 5.1) *Does model extraction work?* **Yes**, model extraction definitively works in that it is possible to approximate a blackbox model from just queries, although not all queries are equally useful.

- (Section 5.2) *Is it possible to extract a model using small number of queries?* **Sometimes**, assuming the attacker has access to some prior knowledge over the distribution and can choose a limited number of high-quality queries that provide task-informative responses.

- (Section 5.3) *Can model extraction be used to reduce the data costs?* **Sometimes**, assuming that OOD responses provide task-informative answers, data costs can be reduced.

- (Section 5.4) *Can model extraction be used both to reduce the data costs and at the same time use small number of queries?* **Sometimes**, if OOD only reveals limited information about the IND behaviour, an ME attacker must either have prior knowledge of the distribution and the ability to sample IND samples or have a very large query budget and traverse the input domain.

- (Section 5.5) *Can model extraction be used to reduce data labeling costs?* **Sometimes**; we find that in practice the victim model provides no additional benefits other than serving as a labeling oracle. For some use cases, there are no more cost-effective sources for obtaining labels; for example, for medical data.

**Experimental Setting.** To answer the questions described above, we evaluate a range of ME attacks on common vision and language benchmarks. We measure the attacker's performance as the accuracy difference between the victim and the attacker on the original task. We define a successful attack as one that produces a model that performs on a task nearly as well as the victim model. Unless stated otherwise, in all our experiments we assume that the attacker has a black-box access to the victim model: the adversary issues a query, and the victim model responds with a vector that indicates the probability of classifying the input in each of the task classes. We note that this setting provides the most information to the adversary. In other words, it advantages the ME adversary.

We define by *baseline attacker* an attacker that has access to a randomly sampled subset of the victim's true training dataset and uses this data only to query the victim and train the attack model. Since ME quality is mainly affected by the quality of the attacker queries, real victim training data represents one of the best query sets for the attacker (compressed datasets *e.g.* Wang et al. (2018) or disjoint IND data can potentially lead to an equally capable extraction set). Throughout our work, we quantify different levels of prior knowledge by the percentage of the data available to the attacker and evaluate the attack accuracy as a function of this measure. We also evaluate ME adversaries that utilize a mixture of IND and OOD data.

**Vision.** For vision tasks, we evaluate the CIFAR-10 dataset (Krizhevsky et al., 2009), for which we follow the setting and training details described by the state-of-the-art DFME attack (Truong et al., 2021). We additionally evaluate the Indoor67 (Quattoni & Torralba, 2009), CUBS200 (Wah et al., 2011) and Caltech256 (Griffin et al., 2007) datasets, and follow the setting and training details described by the Knockoff Nets attack (Orekondy et al., 2019a), one of the strongest ME attacks. Due to space limitation, we provide the Knockoff Nets results in Appendix G, and focus the discussion on CIFAR-10. In all experiments we use the pretrained victim models provided by the authors.

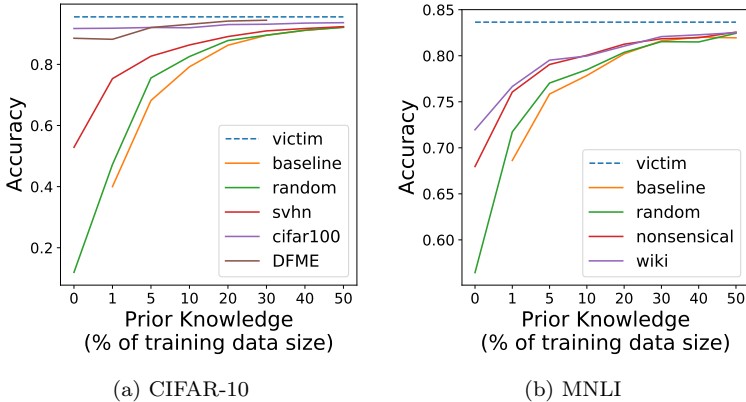

(a) CIFAR-10         (b) MNLI

Figure 2: A comparison between the baseline attacker, which only uses its prior knowledge, and an attacker that can augment its queries with additional queries sampled from other data distributions. We fix the query budget to be the size of the original training set for a fair comparison. Attackers with more prior knowledge do not benefit much by augmenting the query set.

**NLP.** We evaluate the MNLI classification task (Williams et al., 2018), in a standard setting (Krishna et al., 2019). For the victim model, we use the publicly-released, pre-trained 12-layer transformer BERT-base model (Devlin et al., 2018), and fine-tune it for 3 epochs with a learning rate of 0.00003.

## 5.1 Does model extraction work?

We start off by answering the foundational question – does ME work? Our answer to this is **yes**, ME attacks definitely work, and it is indeed possible to approximate a blackbox model from just queries, matching performance to the victim model. As seen in Figure 2, and as evident in the large number of works in this field, there are many settings in which an ME adversary can obtain a model that has a desirable performance over the task of interest.

However, not all queries are equally useful for ME. To demonstrate this, we evaluate an ME attacks on real models that use queries sampled from data distributions that differ from the victim model's training distribution, *i.e.* OOD queries. Here, we consider a number of possible distributions, where some are closer to the true distribution than others. Namely, for CIFAR-10, we use: **Random** - uniformly sampled random images; **Surrogate** - real images sampled from a surrogate (SVHN and CIFAR-100) dataset; **DFME** - synthetic queries generated using the DFME attack. For Indoor67, CUBS200, and Caltech256, we evaluate only the surrogate variant using the ImageNet dataset, as described by Orekondy et al.. Results are presented in Appendix G. For MNLI we use: **Random** - nonsensical sequence of random words built from the WikiText-103 corpus (Merity et al., 2017), sampled letter-by-letter; **Nonsensical** - nonsensical sequence of real words built from the WikiText-103 corpus, sampled word-by-word; **Wiki** - real sentences sampled from the WikiText-103 corpus.

We investigate the case where the query budget is constant and is set to the size of the original training dataset. Higher query budgets can perhaps provide a more successful extraction, but require the cost of collecting and labeling each sample to be even lower to allow a cost effective attack, as discussed in Section 3.2. For each prior knowledge percentage $x\%$, we fill the remaining queries with any of the previously described alternative data sources. We evaluate for up to 50% prior knowledge, because we find that the attacker's query set is dominated by IND samples for higher proportions and is thus less informative. We include the case where all queries are sampled from the query distribution, and the attacker has no (0%) prior knowledge. Note that DFME requires a significantly higher query budget and incurs a high computational cost, therefore we evaluate this setting for a query budget of 20M and with up to 30% prior knowledge.

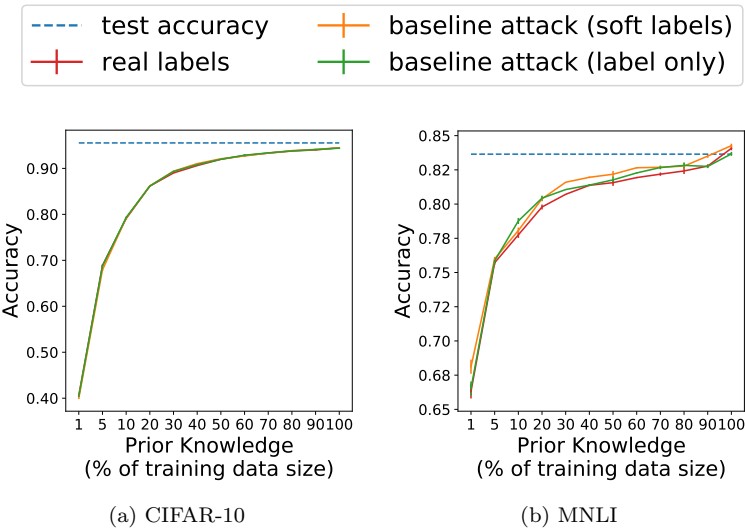

Figure 3: Evaluation of the risk posed by an attacker with some prior knowledge over the true data distribution, using different labeling sources. As can be seen, labels provided by the victim model, either in the richer soft-label setting or in the more restrictive label-only setting, does not provide benefit over the real ground truth labels. This shows that the victim is essentially a labeling oracle.

As can be seen in Figure 2 in the no prior knowledge setting (0%), most attacks were able to obtain a success rate significantly higher than random guess. It is interesting to observe the behaviour of the synthetic queries. The random images or random text based attacks are significantly outperformed by other methods. This comes in contrast to the success of DFME's synthetic samples performance. This contrast suggests that it is a question of query complexity, where in cases where the samples differ from natural samples, a high query budget must be used (DFME's success dramatically reduces when lowering the query budget (Lin et al., 2023)). Additionally, it is interesting to observe the margin between the success of different OOD-based attacks and the success of the baseline attacker. As can be seen, for lower prior knowledge settings, most attacks improve upon the baseline attacker, enabling the attacker to learn additional information using the OOD queries. For example, for CIFAR-10, an attacker with 1% prior knowledge can improve its performance by up to 50%. However, with higher levels of prior knowledge, the attacker gains little benefit from the additional OOD queries. A CIFAR-10 attacker with 50% prior knowledge only increases its success rate by up to 1.4%. In Appendix B we evaluate the effect of additional queries for a given prior knowledge percentage, showing diminishing returns from them.

## 5.2 Can ME be used with only a few queries?

ME seeks to obtain a copy of the victim model while incurring reduced costs compared to training from scratch. In Section 3.2 we note that one way to reduce costs is to limit the interaction with the victim model to the minimum – that is, reduce the overall ME query budget. In this subsection we investigate such reduction and discover that it is indeed **sometimes** possible. In Figure 2 we demonstrate that limiting the query budget for all cases but DFME, still leads to the high attack performance. What is more, even the baseline attacker with only 10% of the data, *i.e.* reducing the query complexity by a factor of 10, manages to develop a model with a relatively high performance, *i.e.* 79.21% compared to 95.54% for CIFAR-10 and 77.85% compared to 83.64% for MNLI; with 50% it approaches the original test accuracy of the model, *i.e.* 92.09% for CIFAR-10 and 81.94% for MNLI. We find that the baseline attacker with more than 50% of the original victim training set can extract the model (see Figure 3). Although we observe this phenomenon consistently in the case above, in our answer to the next question we will show that such good performance heavily depends on the prior IND knowledge or requires OOD data that, as described in Section 4, informs the attacker about the IND performance.

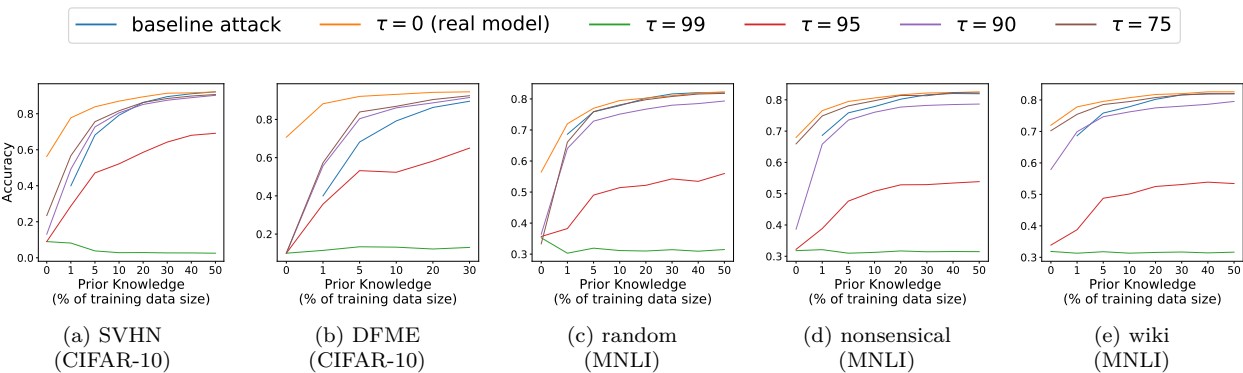

Figure 4: The effect of controlling OOD informativeness with different values of $\tau$ against an attacker that utilizes additional queries. In all cases other than DFME, the attacker adds the size of the training set additional queries, and for DFME adds 20M queries. When comparing the results to the original setting (real model), where the OOD region is unmodified, we can see a clear decrease in the attack accuracy.

## 5.3 Can ME be used to reduce data costs?

A common threat model assumes that IND is scarce and expensive. As such, it is desirable to reduce data collection costs by avoiding, or minimizing the use of IND queries, and prioritizing the use of OOD queries. We find that in some cases ME attacks can indeed do so. More precisely, in Figure 2 we show that OOD queries provide a feasible substitute for IND queries in lower prior knowledge settings. However, we claim that this is only possible because OOD queries inform the adversary about the IND performance *i.e.* IND decision boundaries can be inferred from the OOD ones, either because the distributions are sufficiently similar or because the model optima are predictable. To explicitly estimate if all OOD queries will aid ME performance, we instrument the model with the OOD detector as described in Section 4.3. Our instrumentation, in essence, allows us to control how much information can be derived about the IND from the OOD region.

Per Section 4, an intelligent attacker would have to distinguish between IND and OOD samples to bypass the instrumentation or otherwise waste queries and potentially learn non-existent decision boundaries. A poor attack policy would inevitably learn fake decision boundaries, while a strong attacker would learn to avoid them. Theoretically, in Section 4.1 we discuss that with no prior knowledge, it should be improbable for an attacker to sample from the IND region alone. Furthermore, in Section 4.2 we argue that when samples from both regions are considered, prior knowledge is required to distinguish between the two.

We empirically evaluate this argument using the methodology described in Section 4.3. We show that when the informativeness of the OOD region is deteriorated, OOD queries cannot replace the need for expensive IND queries. We evaluate different threshold values ($\tau$) against an attacker that, in addition to its prior knowledge samples, uses additional $\|D_{train}\|$ queries, *i.e.* the size of the original training dataset. For DFME, we allow that attacker to use 20M additional queries. Different threshold values influence the results by changing the number of additional queries that are predicted by $\mathcal{V}_f$ versus $\mathcal{V}_o$ – the lower the threshold, the better for the attacker. We measure the false-positive rate (FPR) for the different threshold values in Appendix F.3.

The results in Figure 4 clearly demonstrate the described effect. As the attacker has less prior knowledge, it is more reliant on the benefit of the additional queries. As such, it is more impacted by the fake boundaries that control OOD informativeness. The attack accuracy is thus reduced closer to, or below, the accuracy of the baseline, which makes no additional queries. This effect is weaker for some of the settings in our NLP task, specifically the nonsensical and wiki queries. This is due to the similarity between the true data distribution and these query distributions. We discuss this observation in more detail in Appendix H.

### 5.4 Can ME be used both to reduce the data costs and use only a few queries?

Both Q2 and Q3 assessed that ME costs could be reduced by either using substitute data or reducing the number of queries; but can an adversary achieve both? The discussion of Q3 gives a clear answer to this question - no.

We showed that unless the OOD dataset is chosen in such a way as to inform the attacker about the IND behavior, the attacker must either rely on the IND data available to them or utilize a very large query budget. It is impossible to achieve both in this setting. Therefore, our answer to this question would be **sometimes**, as it depends on how much information can be derived about the IND from the OOD region.

### 5.5 Can ME be used to reduce data labeling costs?

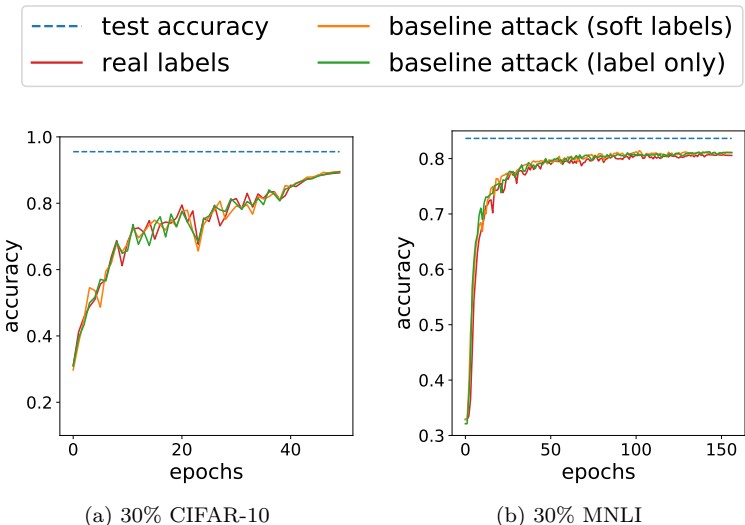

(a) 30% CIFAR-10        (b) 30% MNLI

Figure 5: Comparison between the convergence rate of an attacker that uses the victim's full probability vector output (soft labels), an attacker that utilizes a label-only access to the victim model, and an attacker that uses the real ground truth labels. In all cases the attacker has access to 30% of the true training samples. The attacker does not "learn faster" by attacking the victim model, and only benefits from the victim model when it has little prior knowledge over the true data distribution.

In Q2, Q3, and Q4, we discussed ME assuming a reduction of costs on data collection and the overall number of queries. We concluded that in general, such reduction is unattainable, and OOD queries do not trivially replace IND queries. In this section, we turn to label cost reduction.

Label cost reduction could be achieved in two ways: by reducing the cost of a single query or, alternatively, by increasing the amount of information that a single query provides. Note that an increase in the former does not always mean that overall cost increases, as combined with the increase in the latter, they can cost less overall.

We find that in the classification setting considered in this work, we could not find a method to consistently increase the informativeness of the response, suggesting that the victim model merely serves as a labeling oracle, even in cases where seemingly more information is provided than just a label. To demonstrate this, we revisit our baseline attacker and compare its performance when trained on: 1) *soft-label i.e.* full probability vector; 2) using *label-only*; and finally, 3) using the *real* ground truth labels. Results presented in Figure 3 demonstrate that the attacker's performance is nearly equivalent in all three cases. This phenomenon aligns with the prior literature (Orekondy et al., 2019b). Additionally, we examined each of the cases from a computational perspective and compared their convergence rates. In Figure 5 we present this comparison for the case where the attacker utilizes 30% prior knowledge. We observe that the convergence rate is similar across the three attackers, *i.e.* the attacker does not even "learn faster" by querying the victim model.

In Appendix C we additionally show this for the 5% and 60% prior knowledge settings and observe a similar phenomenon.

We additionally examine the response informativeness effect for attackers who can utilize additional queries for other query distributions. Results, presented in Appendix D, show that for lower prior knowledge settings, and for query distributions that significantly differ from the true distribution, soft-labels provide some benefit over label-only access. However, this effect diminishes as the attacker uses more prior knowledge, or acquires access to a query distribution which is closer to the true distribution.

In light of the finding that the victim model serves as a labeling oracle, how cost-effective is querying the victim? We discussed current annotations costs in Section 3.2, where we suggested that for many use cases, including current ME benchmarks, crowdsourcing annotations may be a cost effective alternative. However, this is not true for all cases, as *e.g.* medical data will most likely have higher data collection and labeling costs. We therefore conclude that it is **sometimes** possible to reduce labeling costs by not running ME and instead finding an alternative labeling provider.

## 6 Conclusion

In this paper we investigate the common assumption that ME attacks are more cost-effective than training a model from scratch. Primarily, it is assumed that OOD data can be utilized for reducing data collection costs, and by using a limited number of queries to the victim model both labeling costs and training costs can be reduced as well. We show that often this is not the case. We demonstrate that the performance of an attacker with a reasonable query budget is bounded by their access to IND data. Augmenting the IND data with data from another distribution relies on informative responses for task-irrelevant queries. We show that decorrelation of OOD from IND responses changes the attacker-victim dynamic, where attacks become much less cost-effective, forcing the attacker to collect IND data. We show that given sufficient IND data, the victim model mainly serves as a labeling oracle, which in turn is not necessarily more cost-effective than online labeling services. We thus conclude that the adversarial goals and incentives of ME attacks should be redefined.

## Acknowledgments

We would like to acknowledge our sponsors, who support our research with financial and in-kind contributions: Amazon, Apple, CIFAR through the Canada CIFAR AI Chair, DARPA through the GARD project, Intel, Meta, NSERC through the Discovery Grant, the Ontario Early Researcher Award, and the Sloan Foundation. Resources used in preparing this research were provided, in part, by the Province of Ontario, the Government of Canada through CIFAR, and companies sponsoring the Vector Institute. We would also like to thank CleverHans lab group members for their feedback.

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

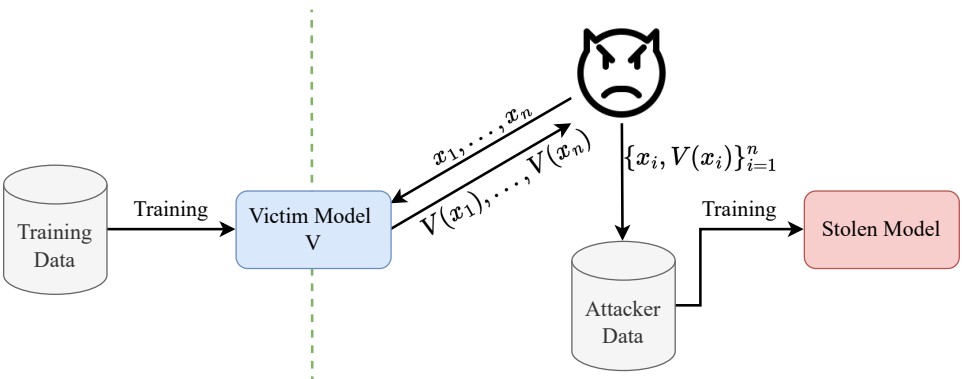

Figure 6: Illustration of model extraction attacks. In this setting, the victim model is trained on some training data that is not accessible by the adversary. The adversary attempts to steal the victim model over query access, in order to obtain an approximate copy of that model with similar performance. Typically, the adversary achieves this by querying the victim model to collect an "attacker dataset". This, in turn, is used to train a stolen, surrogate copy that mimics the victim's behavior.

## A  A note on current literature

Active learning is a branch of semi-supervised learning which focuses on finding query efficient-training regimes, *i.e.* it explores methods that can learn a task with the least number of questions to the oracle. Chandrasekaran et al. formulated that 'the process of model extraction is very similar to active learning' and suggested that improvements in query synthesis active learning should directly translate to model extraction. This relationship works in both directions and implies that greater performance in model extraction directly translated to better active learning regimes.

Some of the current literature reports an ability to extract complex models with a handful of queries *e.g.* Tramèr et al. claims successful extraction of a Multilayer Perceptron with around a thousand queries or Zanella-Beguelin et al. extracts SST-2 BERT with around two thousand queries. This suggests that there exists (very) query-efficient training strategies, with oracles providing labels for otherwise unlabeled data. Yet, in practice, literature in active learning reports less impressive results in these settings. For example, sophisticated state-of-the-art regimes on CIFAR-10 report improvements up to 7% for 5% and up to 5% for 10% of dataset (Yoo & Kweon, 2019; Beck et al., 2021; Yi et al., 2022). Most importantly, the improvements are similarly dominated by the original data access.

Indeed, our paper questions the apparent *free lunch* reported in model extraction literature, and suggests that extraction reported is mostly an artifact of underlying data access, drastically overestimating potency of the attacks. To best understand the underlying attack performance it is imperative to consider the benefit of model extraction for no-data settings and cover them extensively, focusing on the low 0–5% regions.

## B  Effect of surrogate dataset over attack performance

In Section 5.1 we investigated the performance gain of using additional queries drawn from a different data distribution. We evaluated this for a query budget of the size of the original training dataset. This budget represents the maximal query complexity that can still be considered successful. Here, we further extend the results presented in Figure 2 and evaluate the effect of the number of additional queries from various data sources.

Results are presented in Figure 7. We show that, in most cases, larger numbers of additional queries only have a limited effect, which is also very dependent on the quality of the queries and the level of prior knowledge. Adversaries with higher levels of prior knowledge are less affected by the additional queries. The DFME queries, while being out-of-distribution, do perform very well, but at the cost of prohibitively high query budgets.

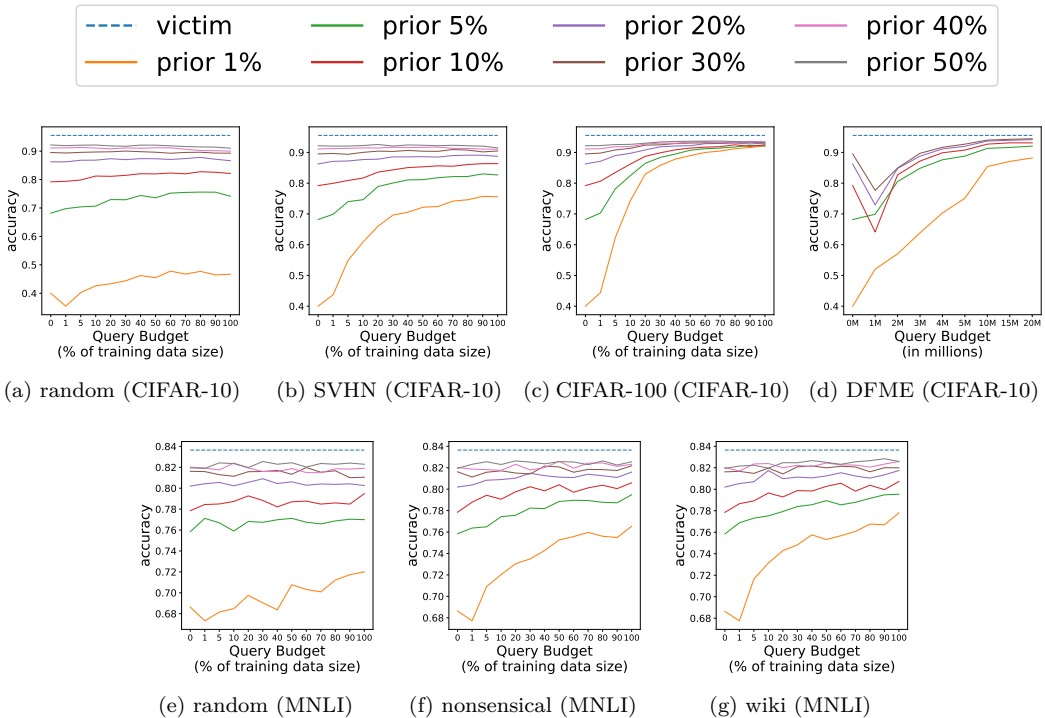

Figure 7: Evaluating the effect of adding different amounts of additional queries for a given level of attacker prior knowledge. Plots (a)-(d) present attacks over the CIFAR-10 victim model and (e)-(f) are for the MNLI victim model. In all plots, a query budget of 0 represents the baseline attack accuracy, as presented in fig. Figure 3. Results show that adding larger amounts of additional queries has a limited effect, which is dependent on the query quality and amount of prior knowledge.

## C  Attack convergence rate comparison

Following the discussion in Section 5.5, we further investigate the actual benefit of attacking the model rather than training from scratch, from the computational perspective. We examine the convergence rate of the attack accuracy. Figure 8 presents a comparison between the convergence rate of the three cases - an attacker that uses the victim's soft labels (*i.e.* full probability vectors), an attacker that has a label-only access to the victim model, and an attacker that uses the real (ground truth) labels. We show that the convergence rate is similar across the three attackers, *i.e.* the attacker does not "learn faster" by querying the victim model. We show that in cases where the attacker has limited prior knowledge over the data distribution, *e.g.* in the 5% case, the attacker does get some benefit from using the victim's soft labels; however, this benefit disappears with increased prior knowledge.

## D  Effect of response informativeness

In Section 5.5 we discussed the role of ME in reducing data labeling costs. We found that, in the setting considered in this work, ME attacks serve as a labeling oracle. To demonstrate this, we showed that when training the baseline attacker with soft-label access to the victim model, i.e,. when the victim model responds to each issued query with a full-probability vector, we did not get any significant improvement over cases where the attacker was trained with label-only access or even the real ground truth labels. In order to further emphasize the limited possible gain an attacker can get by obtaining labeling through an ME attack, we compare the difference in attack performance between soft-label access and label-only access for attackers that can utilize additional queries from other query distribution.

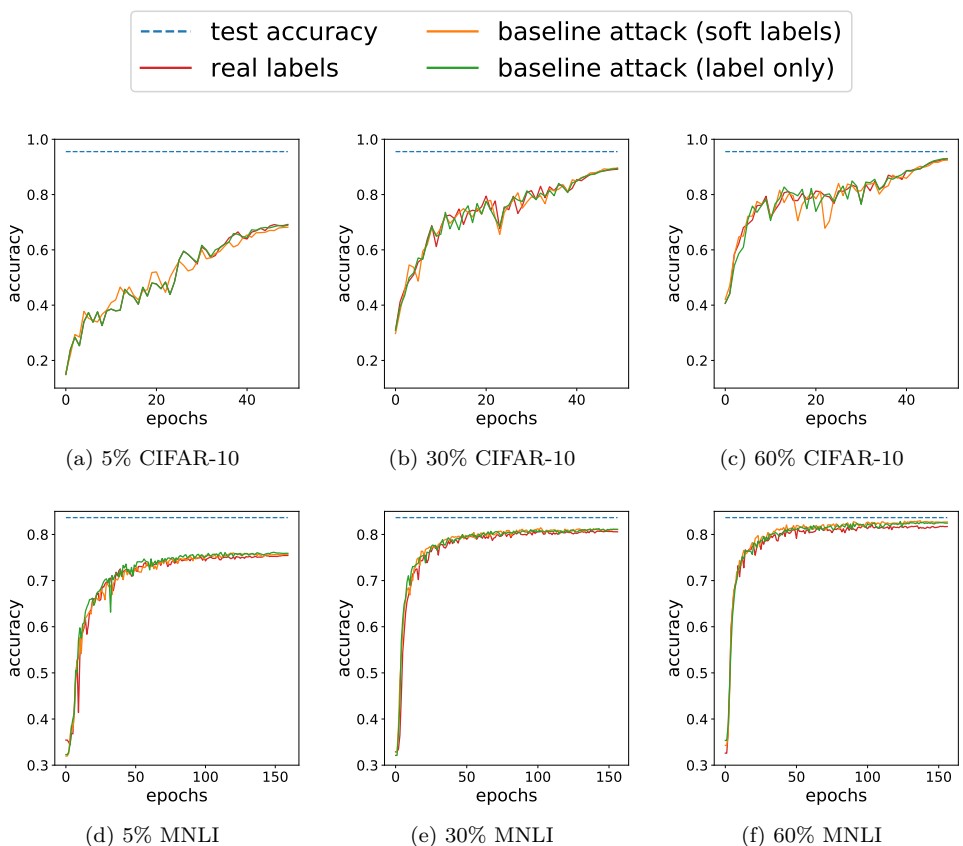

Figure 8: Comparison between the convergence rate of an attacker that uses the victim's full probability vector output (soft labels), an attacker that utilizes a label-only access to the victim model, and an attacker that uses the real ground truth labels. In all cases the attacker has access to 30% of the true training samples. The attacker does not "learn faster" by attacking the victim model, and only benefits from the victim model when it has little prior knowledge over the true data distribution.

Similar to the results presented in Figure 2, we evaluate this for a constant query budget, fixed to the size of the original training set. Due to the high computational cost of DFME, we omit this setting in this evaluation.

The results, presented in Figure 9, demonstrate that soft-labels aid the attacker mainly in the lower prior knowledge settings and for query distributions that are significantly different from the true distribution. In other cases, the victim model serves as a labeling oracle, and the soft labels provide little gain over the label-only access. For example, in the CIFAR-10 case, an attacker that utilizes additional random queries and has access to 1% of prior knowledge can improve it's performance by additional 20.8%, from 26.43% to 47.24% attack accuracy. However, the same attacker with 10% prior knowledge can only improve by 1.25%, from 81.31% to 82.56%, and another attacker with 1% prior knowledge that can utilize SVHN queries can only increase by 7.19%, from 68.14% to 75.33%. In the case of MNLI, the benefit of soft labels is even weaker, due to the similarity between the distributions, as further discusses in Appendix H.

## E   Sampling complexity intuition

In Section 4.1 we discuss the intuition behind the complexity of sampling IND queries with or without prior knowledge over the distribution. Here, we provide a toy example of this complexity for different levels of prior knowledge, modeled by the overlap between the IND and the attacker's query distribution. We then extend intuition for our considered models, and estimate the complexity of sampling IND in this setting.

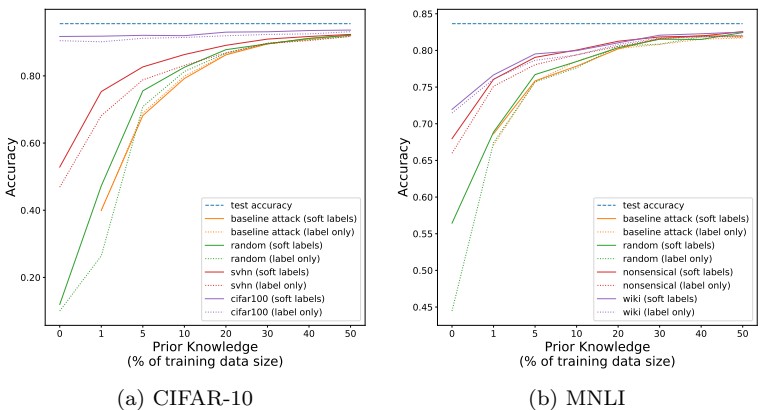

(a) CIFAR-10

(b) MNLI

Figure 9: Comparison between the attack performance in the soft-label setting, i.e., full probability vector, and the label-only setting. Evaluated for the baseline attacker, that utilizes only prior knowledge, and attackers that can utilize additional queries from other distributions (random images, SVHN, and CIFAR-100 in the case of CIFAR-10, and random sentences, nonsensical sentences, and wiki for the MNLI setting). Results demonstrate that soft-labels aid the attacker mainly in the lower prior knowledge settings and for query distributions that are significantly different from the true distribution. In other cases, the victim model serves as a labeling oracle, and the soft labels provide little gain over the label-only access.

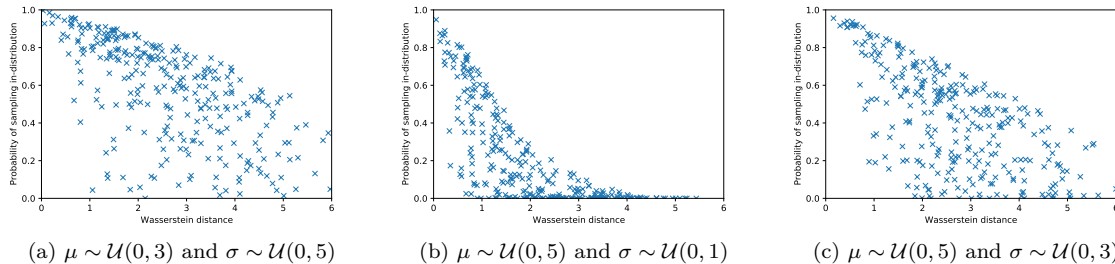

(a) $\mu \sim \mathcal{U}(0,3)$ and $\sigma \sim \mathcal{U}(0,5)$     (b) $\mu \sim \mathcal{U}(0,5)$ and $\sigma \sim \mathcal{U}(0,1)$     (c) $\mu \sim \mathcal{U}(0,5)$ and $\sigma \sim \mathcal{U}(0,3)$

Figure 10: Gaussian overlap as a function of Wasserstein distance.

### E.1 Sampler bias toy example

We explore the relation between a prior knowledge level and the attacker's probability of successfully sampling from the useful domain. We denote the sampler from the useful domain, *i.e.* the in-distribution domain, as $V_s \sim \mathcal{N}(\mu_v, \sigma_v)$, and the attacker's sampler, *i.e.* the distribution from which queries are drawn, as $A_s \sim \mathcal{N}(\mu_a, \sigma_a)$. We model the prior knowledge level as the Wasserstein distance between both distributions.

Figure 10 plots the probability of sampling from the "informative" overlap region as a function of Wasserstein distance between $V_s$ and $A_s$ when $\mu$ and $\sigma$ are sampled uniformly. Small differences in sampling distributions, *i.e.* less prior knowledge, result in a significant reduction in in-distribution sampling probability. This, in turn, results in *wasted queries*, as sampling outside of the overlap is not informative, and in *reduced model capacity*, as the attacker "wastes" capacity on learning the irrelevant OOD region. This holds even in cases where distributions overlap significantly. Note that in practice, with more dimensions, the volume would overlap less, and the useful sampling probability would be even further reduced.

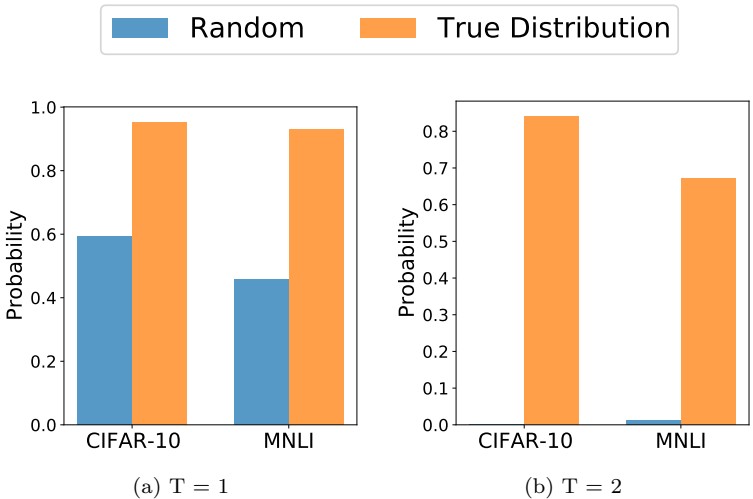

(a) T = 1          (b) T = 2

Figure 11: Estimating the sampling complexity of our real victim models by measuring the percentage of random queries that are predicted by the model with high confidence, *e.g.* above 90%. This is in comparison to the complexity when using queries from the true distribution, sampled from the test set. We evaluate this for both the original model, with a softmax temperature T=1 (a), and for a less confident model, with a high softmax temperature T=2 (b).

### E.2 Sampling complexity in real models

In the previous section, we explore the sampling complexity of an attacker in a toy setting. In this section, we extend this notion to a more realistic scenario of estimating the sampling complexity for our real victim models. For this, we attempt to estimate the in-distribution volume and the complexity of sampling within it by measuring the number of random queries in this volume. Similarly to our OOD control module, described in detail in Appendix F, we define a query as in-distribution, *i.e.* inside the volume, by observing the model's confidence in predicting this query. High confidence queries, above some predefined confidence threshold, are considered to be in-distribution. Therefore, for the task of volume estimation, we measure the percentage of random queries that are above this threshold. In this section, we use a threshold value of 90.

In Figure 11 (a) we present our estimation in both victim models. As can be seen, only 59% of the random queries are sampled from within the volume in the case of the CIFAR-10 model, and only 45% in the case of the MNLI model. We additionally compare this to the sampling complexity when using real in-distribution data, by measuring the percentage of samples from the test set that are predicted by the model with a high confidence. In this case, where we have significant prior knowledge over the distribution, the sampling complexity is drastically decreased.

As our estimation dependents on the model confidence, it is also interesting to observe how the complexity changes when the model becomes less confident. For this reason, in Figure 11 (b) we perform the same evaluation, however, we decrease the victim model's confidence by increasing its softmax temperature from 1 to 2. This change results in a less confident model in general and, as evident from the results, it is nearly impossible to sample from within the distribution without any prior knowledge: only 0.00001% for CIFAR10 and 0.01 for MNLI.

## F   Controlling OOD informativeness - full implementation details

In this section, we elaborate on the implementation details behind our methodology for evaluating the effect of limiting the utility of OOD queries, described shortly in Section 4.3. Given the original victim model $\mathcal{V}_o$, we create a hybrid victim model $\mathcal{V}_h$ by combining the original victim model $\mathcal{V}_o$ with an additional module with different, or additional, decision boundaries $\mathcal{V}_f$. For each query $x$, we apply a decision rule $R$ over $x$, to determine which of the two modules, $\mathcal{V}_o$ or $\mathcal{V}_f$, should be used for predicting this query.

The decision rule $R$ is implemented by applying a threshold value $\tau$ over the prediction confidence of $\mathcal{V}_o$. If a query $x$ has a low prediction confidence, we define it as an OOD query and return the prediction $\mathcal{V}_f(x)$. Otherwise, we define it as IND and return $\mathcal{V}_o(x)$. We calibrate $R$ by increasing the model's Softmax temperature to 2, as discussed in more detail in Appendix F.2. Thresholding the prediction confidence serves as a naive approximation for OOD detection, and using more sophisticated OOD detection methods will only increase the effect of modifying the OOD behaviour. As such, the exploration of different decision rules is out of scope for this work.

As for the fake model $\mathcal{V}_f$, we implement it by fitting a Gaussian Mixture Model (GMM) for each class of the training data. For each class $c \in \{1, 2, ..., C\}$, where $C$ is the number of classes, we sample $S = 5000$ training data points labeled as class $c$, $i.e.$ $\{(x_1, c), (x_2, c), ..., (x_S, c)\}$. We then use the victim model $\mathcal{V}_o$ to compute the predictions of this set, $i.e.$ the logits. We fit a GMM to this set of logits $\{\tilde{y_1}, \tilde{y_2}, ..., \tilde{y_S}\}$, where $\tilde{y_i} = \mathcal{V}_o(x_i)$.

For each class, we create $m = 5$ anchor points $\mathcal{A}_c^1, \ldots, \mathcal{A}_c^m$ that will be used to "assign" queries for this class. For this, we first compute the feature representations of the data samples used for fitting the class GMM, $i.e.$ $\{(x_1, c), (x_2, c), \ldots, (x_S, c)\}$, using some feature extractor $\phi$. Then, we cluster these feature vectors $\{\phi(x_1), \ldots, \phi(x_S)\}$ into $m$ clusters using the Kmeans clustering algorithm. We define the anchor points to be the centroids of each cluster. For vision tasks, we use a pre-trained ResNet34 for the feature extractor $\phi$; for NLP tasks we use a pre-trained BERT-base model.

At last, we sample $m$ permutations $\pi_j : C \to C$, for $j \in [m]$. The permutations would ensure that no query would be predicted using its "real" assigned class, therefore avoiding tail-of-distribution samples classified as OOD samples ($i.e.$ false-positives samples), to be correctly labeled and leak information about true IND behaviour.

For a given query sample $x$, we compute its feature representation $\phi(x)$ and find the nearest anchor point $\mathcal{A}_i^j$, in terms of the $L_2$ distance between $\phi(x)$ and all $C \times m$ anchor points. The anchor point $\mathcal{A}_i^j$ represent the $j^{th}$ anchor point of the $i^{th}$ class. We then permute the assigned class $i$ using the $j^{th}$ permutation, $i.e.$ $i' = \pi_j[i]$, and sample a "fake" logit $\tilde{y}$ from the GMM we fitted earlier to class $i'$. This value is returned as the complex model prediction, $\mathcal{V}_f(x) = \tilde{y}$.

This construction requires the evaluation of a subset of the training data used for fitting the GMMs. The evaluation, fitting, and clustering process are done only once; hence, it is relatively computationally inexpensive. During inference, each query directed to the fake model $\mathcal{V}_f$ adds some computational cost of computing its feature representation, and performing a nearest neighbor search. However, this is only true for OOD samples and false-positive IND samples. Most legitimate users' queries, $i.e.$ the true-positive queries, are completely unaffected by the introduction of this additional module.

It is important to note that while our method decreases the utility of existing ME attacks, as shown in our results, we do not present it as a defense mechanism but rather as a method for evaluating our hypothesis. An efficient defense mechanism can be designed based on these principles, with some additional effort. However,

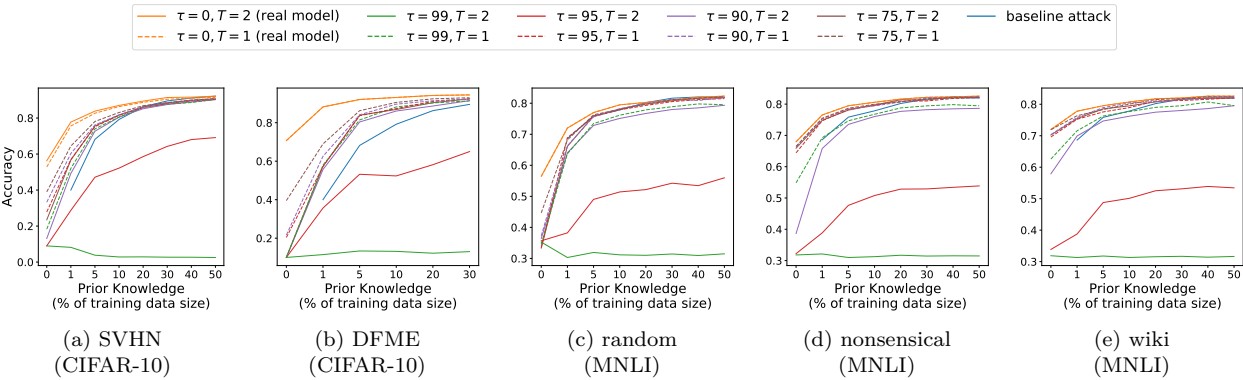

Figure 12: As our OOD detection rule is distinguishing between in and out of distribution queries based on the model confidence, we evaluate the effect of decreasing the model's confidence by increasing the softmax temperature T from 1 to 2. Results shown in this plot show that this indeed even further lowers the benefit the attacker gains by utilizing OOD queries.

this is out of scope for our paper. We propose a method for measuring the reliance of ME attacks on the informativeness of OOD regions, as part of a bigger discussion over the role of prior knowledge and the common methodology of ME attacks, and do not investigate the utilization of it as part of any defense mechanism.

## F.1  Effect of anchor points and permutations

The design of our proposed OOD module $\mathcal{V}_f$ satisfies two main objectives. The first, and most important, is to avoid leaking real IND behaviour by the predictions made by $\mathcal{V}_f$. The second, was to lure the attacker to waste effort and capacity in learning fake and more complex decision boundaries, and therefore reducing further it's performance over the real IND decision boundaries.

To obtain the first goal, we introduced the permutation mechanism. As mentioned before, some tail-of-distribution samples are falsely classified as OOD samples, *e.g.*false-positives. As these samples share the same distribution as the anchor points, when searching for the nearest neighbor anchor point, they would be correctly assigned to their real class. When no permutation is applied, this will result in a prediction that resembles the original prediction given by $\mathcal{V}_o$, and therefore leaks information. To avoid this, the matching anchor point should direct the sample to a different class by using some class-level permutation.

To obtain the second goal, we introduce multiple new decision spaces by using multiple permutations per class. As a result, two samples that were initially assigned to one class can now be directed to two different classes, placing a new decision boundary between them. For this, we use multiple ($m = 5$) anchor points per class, and each anchor point is coupled with a different permutation. Therefore, two samples $x_1$ and $x_2$ that were assigned to two anchor points related to class $i$: $\mathcal{A}_i^1$ and $\mathcal{A}_i^2$, will now we reassigned to two different classes. $x_1$ would be predicted using the GMM fitted for class $j = \pi_1(i)$ and $x_2$ would be predicted using the GMM fitted for class $k = \pi_2(i)$.

To better demonstrate the effect of using multiple permutations and anchor points, we provide an ablation study in Figure 20. We compare the performance of our OOD module in 4 different settings: (i) using one anchor point per class and no permutations (ii) using one anchor point per class with class-level permutation (iii) using 5 anchor points per class with the same permutation shared between all anchor points (iv) our proposed method - 5 anchor points per class with 5 different permutations. These would be denoted as "1 anchor, no perm", "1 anchor, 1 perm", "5 anchors, 1 perm", "5 anchors, 5 perms", respectively. The biggest effect can be attributed to the simple addition of permutations; however, it can be further emphasized by incorporating the additional anchor points and multiple permutations.

### F.2 Softmax temperature influence

In order to calibrate $R$ to better distinguish between IND and OOD samples, we increase the model's Softmax temperature to 2. By doing so, we force the model to be less confident, which results in more queries being predicted by $\mathcal{V}_f$ and not $\mathcal{V}_o$. As shown in Figure 11, a temperature value of 2 indeed detects most of the OOD queries. In Figure 12 we demonstrate the effect of the temperature value over the behaviour of our controlled informativness experiment and show that a better calibrated decision rule indeed further lowers the benefit the attacker gains by utilizing OOD queries.

It is important to note that this also increases the FPR of the IND samples, *i.e.* the percentage of IND samples predicted by $\mathcal{V}_f$ instead of $\mathcal{V}_o$. A higher FPR results in a decline in model test accuracy. However, as shown in Table 1, and described in more detail in Appendix F.3, this decrease is not linearly dependent on the exact FPR.

### F.3 Impact of $\tau$

The value of the confidence threshold $\tau$ determines the utility – extraction difficulty trade-off. In Section 4.3 we evaluate our OOD control mechanism on different values of $\tau$, and observe the effect each has over the attack performance. In order to verify that we are not trivially rejecting all inputs, we additionally measure the false-positive rate (FPR) for the different threshold values. In Table 1, we detail the relation between the threshold value, the FPR, and the effect on the victim's test accuracy. For lower $\tau$ values, most queries are predicted using the original $\mathcal{V}_o$, *i.e.* the victim model's utility is barely harmed since most queries are answered by the real model. In this case, the attacker's performance is barely affected, as it can still get informative responses by issuing OOD queries. As $\tau$ increases, more queries– including some IND ones– are predicted by the "fake" model $\mathcal{V}_f$. This makes it more difficult for an attacker to infer which decision boundaries are IND. We discussed the underlying complexity in detail in Sections 4.1 and 4.2. As can be seen, the attack accuracy dramatically decreased, which verifies that the OOD behaviour leaks almost no information about the IND behaviour. To further demonstrate this, we present in Figure 13 a comparison between the labels predicted by $\mathcal{V}_o$ and those predicted by $\mathcal{V}_f$ for actual OOD queries (*i.e.* true positives) as well as tail-of-distribution IND samples that were detected as OOD (*i.e.* false positives). We show this comparison for an attacker that utilizes additional SVHN queries, 30% prior knowledge, and for a threshold of $\tau = 95$. It is clear that, although the "fake" labels are heavily biased towards one class for the additional queries, in both cases, they are uncorrelated with victim models' predicted labels.

| | CIFAR-10 | | | | MNLI | | | |
| | Temp. 1 | | Temp. 2 | | Temp. 1 | | Temp. 2 | |
| $\tau$ | FPR | Acc | FPR | Acc | FPR | Acc | FPR | Acc |
|---|---|---|---|---|---|---|---|---|
| 0 | 0 | 95.5 | 0 | 95.5 | 0 | 83.6 | 0 | 83.6 |
| 75 | 2.5 | 94.5 | 7.4 | 91.4 | 3.5 | 83.2 | 9.1 | 81.9 |
| 90 | 4.7 | 93.2 | 15.9 | 84.1 | 7.1 | 82.4 | 32.8 | 72.8 |
| 95 | 6.5 | 92.0 | 40.5 | 60.4 | 10.4 | 81.5 | 71.1 | 49.6 |
| 99 | 10.6 | 88.92 | 99.7 | 2.32 | 27.7 | 75.2 | 100 | 32.1 |

Table 1: The threshold value $\tau$ determines which samples are marked as in-distribution, predicted by the original model $\mathcal{V}_o$, and which are treated as OOD and predicted by $\mathcal{V}_f$. The higher the threshold, the higher the false positive rate is. This, in turn, has a negative effect over the teacher test accuracy. This also implies that higher threshold values result in a higher true positive rate. The Softmax temperature (denoted as "Temp." in the table) also affects the FPR and can be used to better calibrate the OOD detection component.

### F.4 On learning of fake boundaries

Although modern learning is an inherently stochastic process, learning with standard tools such as SGD has a bias towards structured solutions (Soudry et al., 2018; Mousavi-Hosseini et al., 2022). For the same

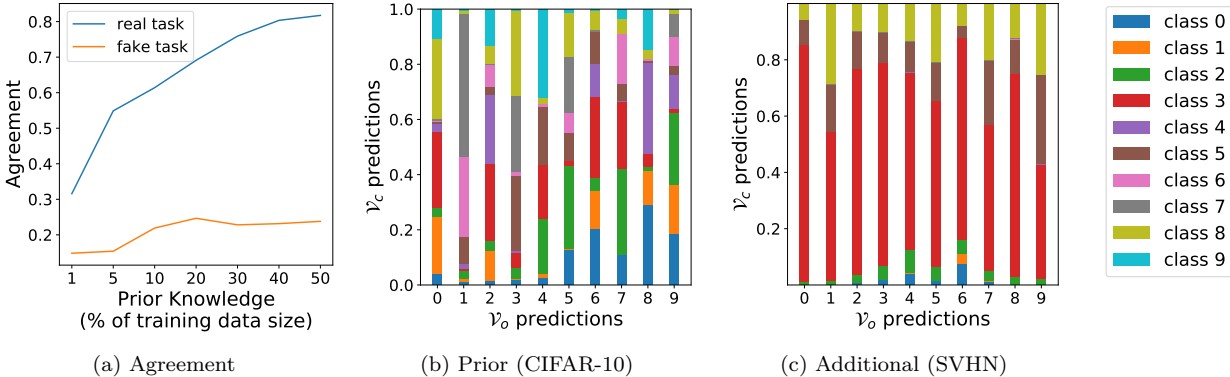

|   |   |   |
|---|---|---|
| (a) Agreement | (b) Prior (CIFAR-10) | (c) Additional (SVHN) |

Figure 13: (a) The agreement between the attacker model and the victim model, for the CIFAR-10 task and an attacker utilizing SVHN additional queries. The agreement is separated into the real task ($\mathcal{V}_o$) and the fake task ($\mathcal{V}_f$). We can see that fake task agreement is higher than random guessing (10%), which implies that the attacker was able to learn the fake, irrelevant task, and waste some capacity.

(b)-(c): Comparison between the labels predicted by $\mathcal{V}_o$ (x-axis) and the labels predicted by $\mathcal{V}_f$ (y-axis), for an attacker that utilizes additional SVHN queries, 30% prior knowledge, and for a threshold of $\tau = 95$. Each bar $i$ demonstrates the distribution of the "new" fake labels that would have been predicted by $\mathcal{V}_o$ as class $i$. The y-axis is normalized to show the percentage of samples from each class. Although the fake models' predictions are biased towards one class for the additional queries, in both cases they are un-correlated with the victim models' predictions, and therefore present new decision boundaries for the OOD queries.

model parameter budget, SGD learns the smoothest boundaries first and only gets to the other boundaries if the capacity permits (Ben Arous et al., 2021). This has real practical implications on the fake regions that we add to the model. Namely, we can not add decision boundaries that are more complex than the real task, since SGD learns to ignore them in light of the real decisions, limiting the extent to which we can add arbitrary complexity into the models.

In Section 4 we have discussed the query budget and model capacity that the attacker must spend if it can not distinguish between the task-related (IND) and unrelated (OOD) queries. To verify, we explicitly check that the attacker indeed learned both the task related and unrelated knowledge, and did not "ignore" the predictions made by $\mathcal{V}_f$ due to the SGD bias described above.

For this reason, we investigate the agreement between the attacker model and both $\mathcal{V}_o$ and $\mathcal{V}_f$, for the SVHN additional queries case. We separate the agreement of the samples predicted by $\mathcal{V}_o$ and the samples predicted by $\mathcal{V}_f$. Figure 13 demonstrates that the attacker model agreement with $\mathcal{V}_f$ is higher than chance level, which is 10% in this case (for the 10 CIFAR-10 classes). This proves that the attacker learned both the real and the fake task and, as such, wasted capacity on learning irrelevant decision boundaries.

## G  Knockoff Nets Evaluation

In addition to the CIFAR-10 and MNLI datasets, we evaluate our main experiments on the Indoor67 (Quattoni & Torralba, 2009), CUBS200 (Wah et al., 2011) and Caltech256 (Griffin et al., 2007) datasets. For this, we follow the setting and training details considered by the Knockoff Nets attack (Orekondy et al., 2019a), and use the pre-trained victim models provided by the authors. Orekondy et al. provides two strategies for sampling queries (i) random - in which the queries are sampled uniformly at random from some query distribution (ii) adaptive - in which the queries are sampled according to a learned policy $\pi$. We note that there is no official implementation for the adaptive strategy, and thus we reimplement the method with the details provided by Orekondy et al.. We simplify the hierarchy to be one-level deep, and omit the coarse-to-

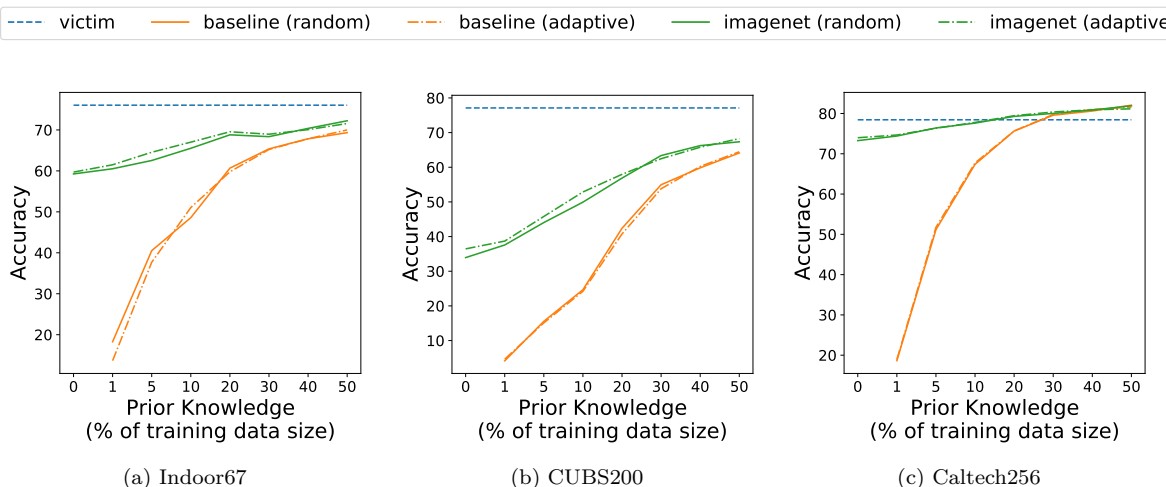

Figure 14: We evaluate the effect of augmenting the attacker's queries with additional queries sampled from the ImageNet dataset, in comparison to our baseline attacker, which only use its prior knowledge. We fix the query budget to be the size of the original training set to provide a fair comparison between the different attackers. It can be seen that, as the attacker has more prior knowledge over the true distribution, it does not gain much benefit by augmenting the query set. Since the ImageNet dataset does share some distributional similarity with the true training data distributions, we observe a significant improvement from utilizing it in the lower prior knowledge settings.

fine label hierarchy used to supplement the policy. We observe similar performance to the original results in (Orekondy et al., 2019a).

For the attacker model, we use a ImageNet pretrained ResNet-34 architecture. We refer the reader to Orekondy et al. for full training details.

Similar to the evaluations in Section 5.1, we show in Figure 14 that ME indeed works, and the model can be approximated with blackbox query access. To better visualize this, we compare the performance of a baseline attacker, that only uses its prior knowledge over the distribution, with that of the OOD-based attacker. Here, the baseline attacker is assumed to have access to either a randomly or adaptively sampled subset of the victim's true training dataset. Following Orekondy et al. we use ImageNet (Deng et al., 2009) as the surrogate (additional) dataset, from which we either sample randomly or using the adaptive strategy. We follow the setting described in Section 5.1, and fix the query budget to be the size of the original training dataset. The results, presented in Figure 14 align with the results in Section 5.1, as they show that as the attacker has more prior knowledge, it benefits less from utilizing additional queries. We do observe that in the lower prior knowledge settings, the ImageNet additional queries do provide a significant improvement, especially for the Indoor67 and Caltech256 datasets. We hypothesize that this is due to a high similarity between the distributions. This also aligns with the main findings of our paper.

Figure 14 additionally answers the questions described in Section 5.2 and Section 5.3. Namely, ME can be used with only a few queries, as the attack is succesfull both when bounding the query budget to the size of the original training set, and when using just a smaller fraction of IND samples. Moreover, the results also show that the use of ImageNet queries as OOD data is indeed effective and can help reduce the data costs involved with using expensive IND samples.

However, as we discussed in Section 4 and further evaluated in Section 5.4, the answers to the previous questions depends on the implicit assumption that the IND decision boundaries can be inferred from the OOD ones. We evaluate this assumption in this case study as well, following the methodology described in Section 4.3. We set the model's softmax temperature to 2, and use $\|D_{train}\|$ additional queries.

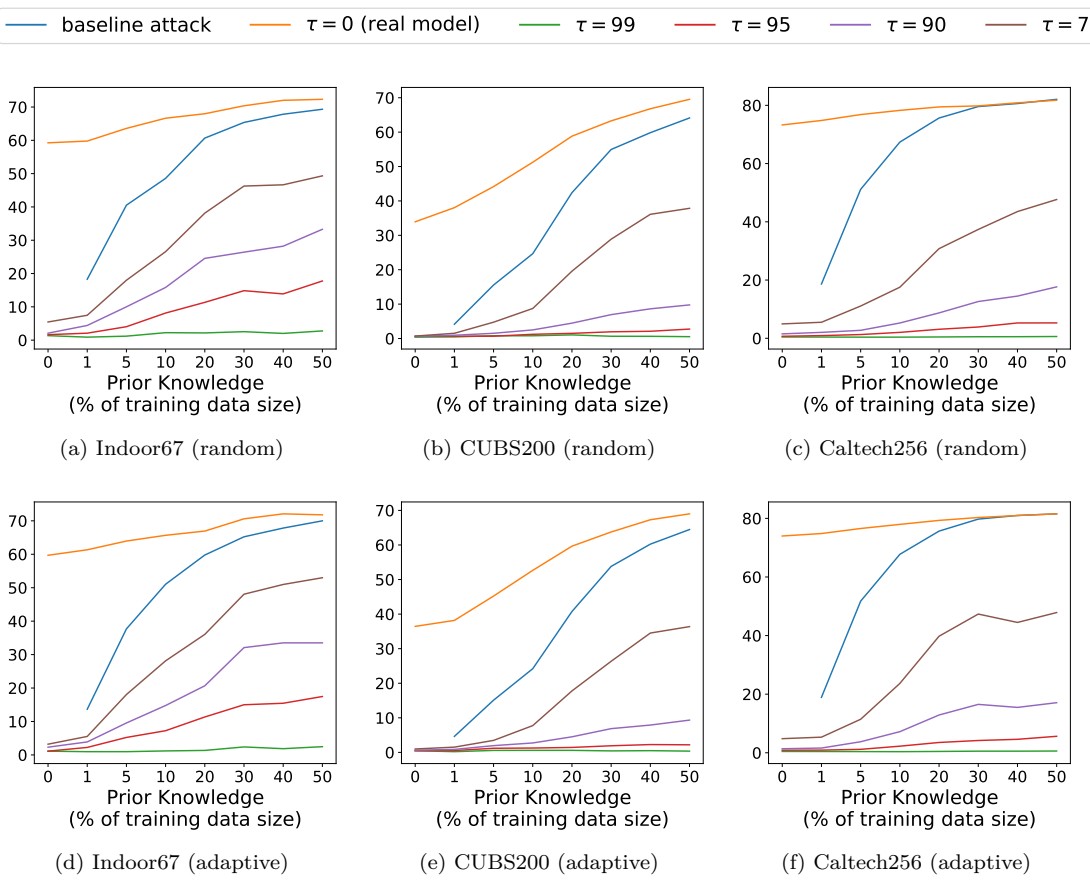

Figure 15: The effect of limiting the OOD informativeness with different values of $\tau$ against an attacker that utilizes $\|D_{train}\|$ additional ImageNet queries. When comparing the results to the original setting (real model), where the OOD region is unmodified, we can see a clear decrease in the attack accuracy.

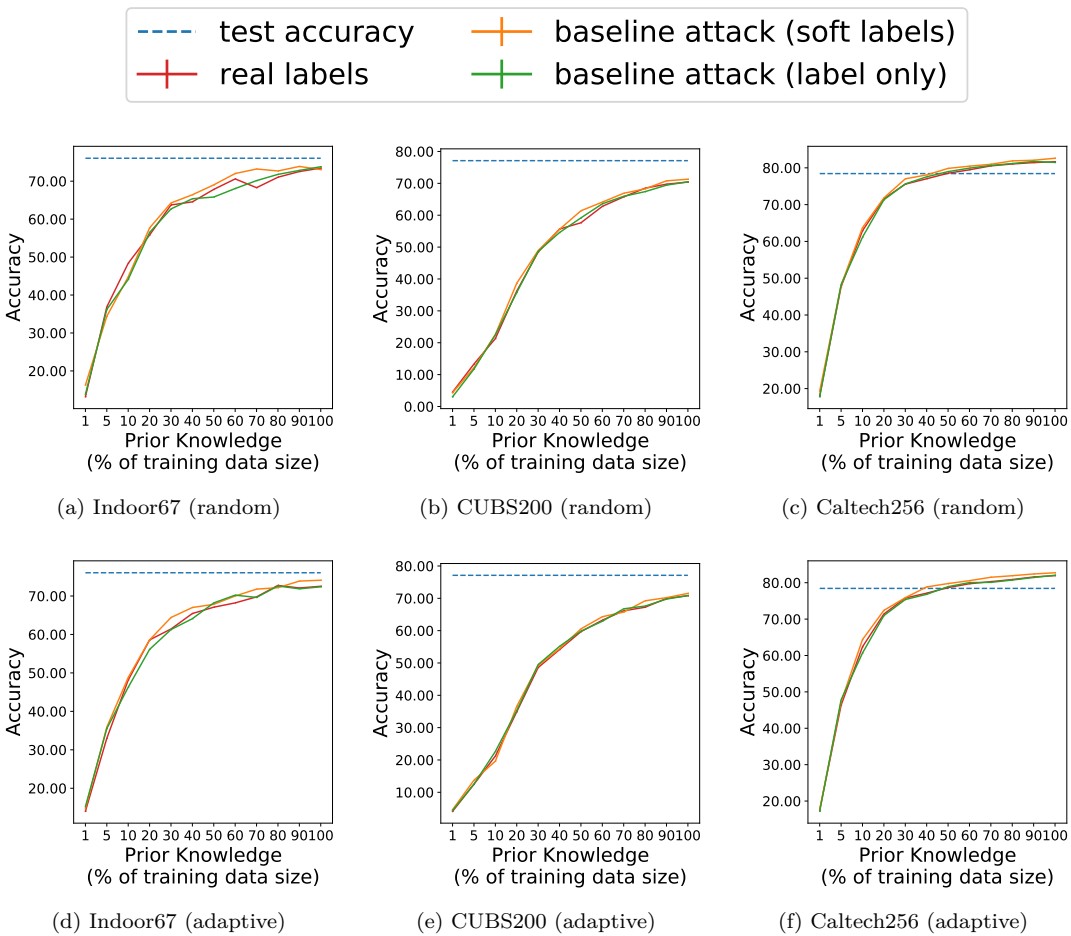

Figure 16: Empirical evaluation of the risk posed by an attacker with some prior knowledge over the true data distribution. The prior knowledge is expressed as access to a percentage of the true training set. In most cases, an attacker with more then 50% access can nearly fully extract the model, however, in this case, the extraction is not more efficient than training from scratch. The attacker does not gain much by querying the victim model versus using the real labels, which also seems to be equivalent to a label-only access to the victim model. This shows that, other than performing as a labeling oracle, the extraction is meaningless in this case.

Figure 15 demonstrates that limiting the informativeness of OOD queries indeed decreases the benefit the attacker gained by utilizing additional ImageNet queries, even to the point of performing worse than the baseline in the higher prior knowledge settings. This again aligns with our findings in Section 5.4 and suggests that the attacker can not acheive both goals, and in the absence of prior knowedlge, corresponding to higher data collection costs, a high query budget must be used.

Aligning with the findings reported in Section 5.5, we present in Figure 16 a comparison between the attack performance using soft-label access to the victim model to that of an attacker with access to the real (ground truth) labels or label-only access to the victim model. This comparison provides the same evidence as described in Section 5.5. It demonstrates that attacking the victim model is merely using the victim model as a labeling oracle in the absence of access to the real labels, and the attacker does not gain much additional benefit. This phenomena was also observed by Orekondy et al..

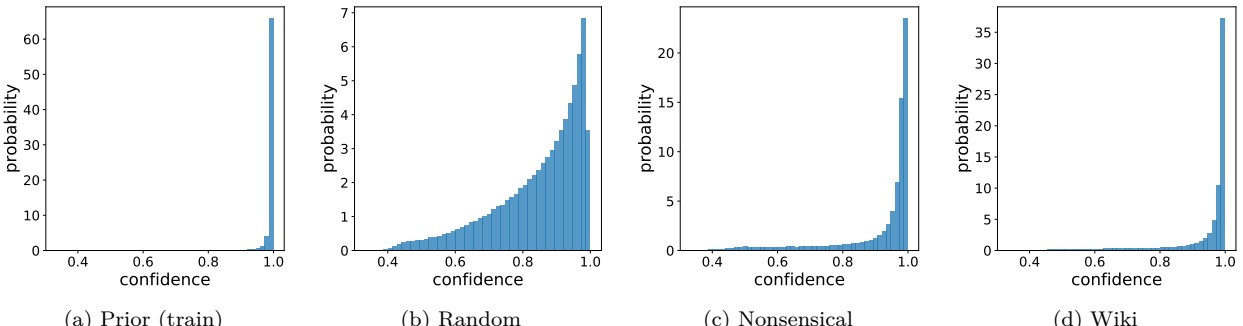

(a) Prior (train)  (b) Random  (c) Nonsensical  (d) Wiki

Figure 17: We compare the confidence values of the victim model over the true data distribution versus the different types of out-of-distribution queries. Results show that the model tends to be highly confident for both in-distribution as well as for some of the out-of-distribution queries, making it harder for the OOD component to make any impact for lower threshold values. This result can explain the surprising success of using random nonsensical queries in the language domain versus the poor results of using random queries in the vision domain.

## H   NLP distribution similarity

The results presented in Figure 4 show that the effect of limiting the leakage from OOD behaviour to IND behaviour is weaker for some of the settings in our NLP task, specifically the nonsensical and wiki queries. This might seem confusing. The wiki queries are broadly equivalent to the surrogate dataset queries that are commonly used in the vision domain, however they exhibit a different response in our evaluation. As both random images and random nonsensical sentences have no real meaning, one might expect similar behaviour between them. In the past, nonsensical queries were even used as a benchmark under the name of random queries *e.g.* by Truong et al..

This difference between the vision and NLP domains is due to the similarity between the true data distribution and the aforementioned query distribution. Although the queries are drawn from either a random nonsensical distribution in the *nonsensical* case, or a from a different corpus in the *wiki* case, we observe that many common words are shared between all three distributions. This results in similar model behaviour between the OOD queries and the true data. It can explain the success of the queries in this domain, which comes in significant contrast to the lack of success of random or some surrogate queries in the vision domain, where the input space is a continuous pixel space rather than a (relatively small) discrete dictionary of words. The similarity between distributions results in high confidence values for both in-distribution and out-of-distribution queries, making it difficult to separate them and apply our method. We show the confidence distribution for each query type in Figure 17. In the case of the random queries, where each letter is sampled, and the sentences are not composed of real words, we can observe a significantly lower gain from the additional queries, which is more in line with the findings from the vision domain.

## I   Task and model complexity

In this section, we investigate the relationship between task and model complexity, as well as the query complexity of model extraction attacks. In Figure 18 we observe that for simpler tasks, such as classification over the SVHN and SST-2 datasets, even an adversary with little prior knowledge can successfully extract the model, *e.g.* 5% data access. This is in contrast to the results we demonstrated for the CIFAR-10 and MNLI datasets, which are significantly harder tasks to learn.

We additionally investigate the effect of the size of the victim model architecture. In addition to the ResNet-34-8x CIFAR-10 victim model evaluated so far, we trained two CIFAR-10 victim models: a larger ResNet-50-8x and a smaller ResNet-18-8x. Note that we trained these models ourselves, while for the ResNet-34-8x architecture, we used the pre-trained model by Truong et al.. The results, presented in Figure 19, show that

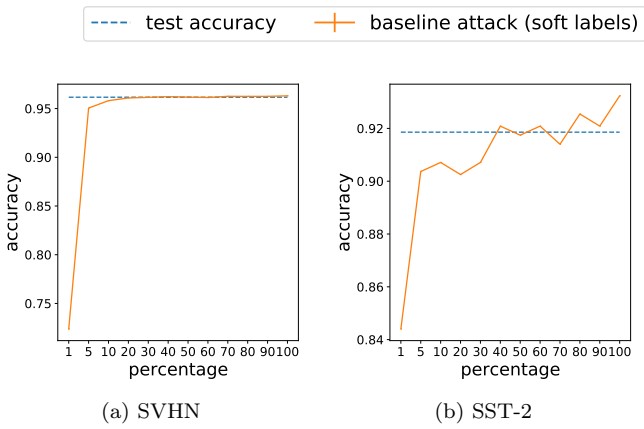

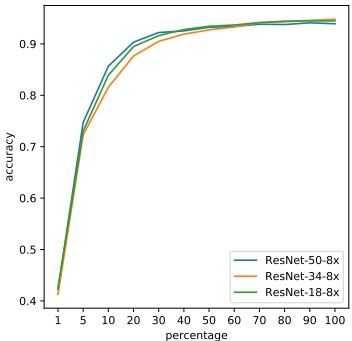

(a) SVHN    (b) SST-2

Figure 18: The baseline attacker is able to successfully extract some victim models very easily, with as little as 5% of the training data. We hypothesize that this is due to the inherent linearity of these datasets.

Figure 19: We investigate whether the victim model size has any impact on the attack success. We observe no impact when increasing (ResNet-50-8x) or decreasing (ResNet-18-8x) the model size. All models are incredibly over-parameterized and, therefore, the differences between the models are insignificant in terms of the attack complexity.

this has minimal impact on the attack success. Lack of differences here can be explained by the fact that all three models share similar accuracy - 94.22% for ResNet-18-8x, 95.54% for ResNet-34-8x, and 93.72% for ResNet-50-8x.

As previously noted, the models in practice mainly serve as labeling oracles, meaning the difference in the attack performance is connected to the model accuracy. Having said that, it is worth mentioning that we have not performed a thorough hyperparameter search in the training of the models. This should not affect the validity of the results and should only cause a slightly lower accuracy.

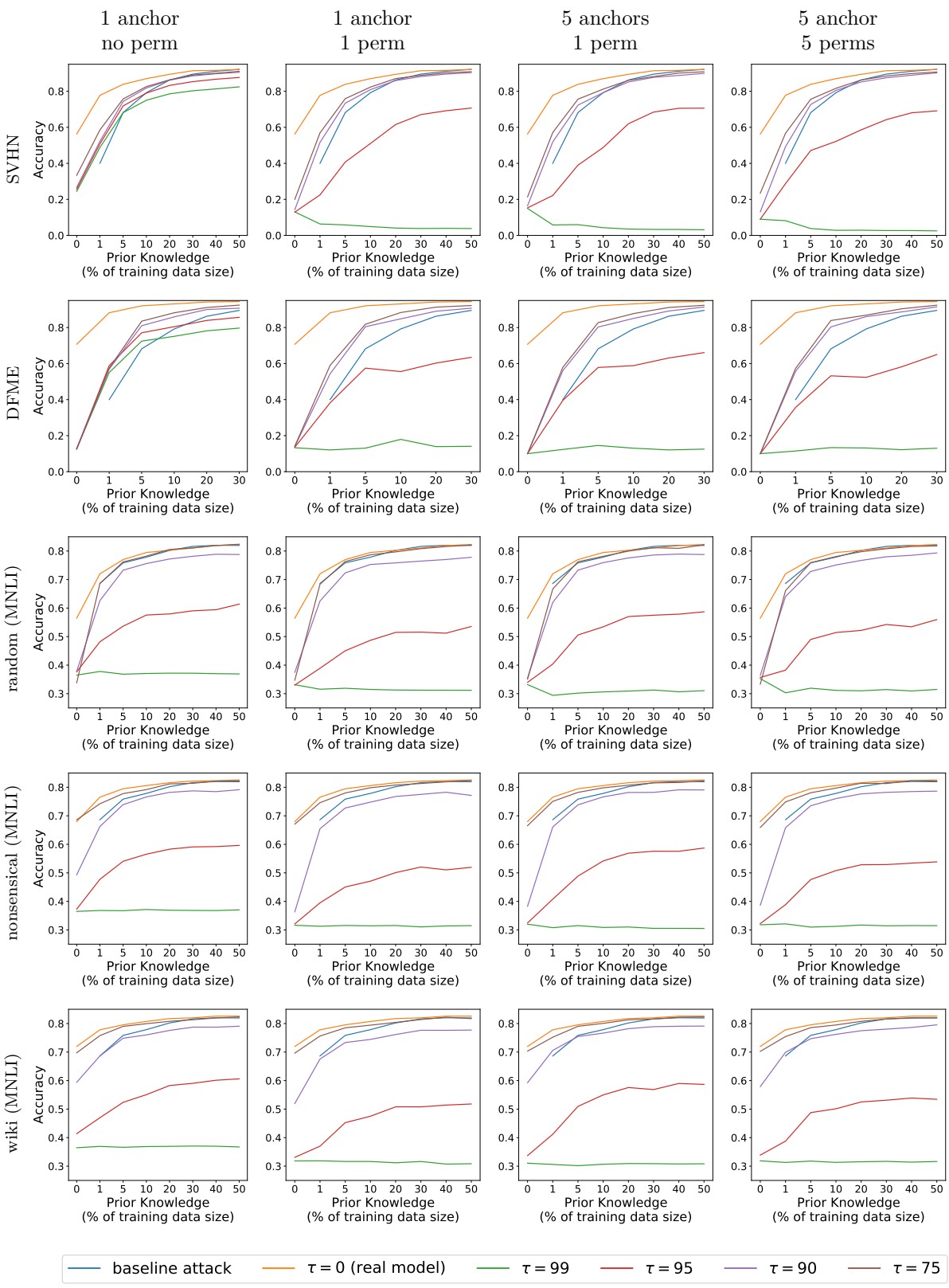

Figure 20: An ablation study of the use of multiple anchor points and permutations in the instrumentation of the OOD module. Comparing between 4 settings shows that indeed our proposed method ("5 anchors, 5 perms") results with the biggest degradation of the attacker's performance even when given access to higher levels of prior knowledge.

