# OpenReview forum: "Beyond Labeling Oracles -  What does it mean to steal ML models?"
_TMLR — Accepted by TMLR_

### Review · Reviewer_8vE9 · 2024-08-14

**Summary Of Contributions:**

The research challenges the common assumption that model extraction is cost-effective because it allows attackers to bypass the expensive process of data collection and labeling. The success and efficiency of these attacks are heavily dependent on the attacker’s prior knowledge and access to in-distribution data. And conclude that the adversarial goals and incentives of ME attacks should be redefined.

**Audience:**

Yes

**Broader Impact Concerns:**

None.

**Claims And Evidence:**

Yes

**Requested Changes:**

Refer weaknesses.

**Strengths And Weaknesses:**

Strengths:

1. The paper argues that the practical threat posed by ME attacks is often exaggerated. The costs borne by the attacker to successfully extract a model are frequently comparable to those incurred by the victim. This is because without prior knowledge or access to in-distribution data, the query complexity and associated costs for the attacker increase significantly.

2. The introduction of the OOD informativeness control method is a novel approach to studying the impact of query distribution on ME attacks.

3. The authors suggest that the community needs to redefine the adversarial goals of ME attacks because current evaluation methods often misinterpret ME performance through theory analysis and extensive experiments.

Weaknesses:

1. In the context of ME, ME would be more cost-effective than training a model from scratch. In practice, for a deep learning model, the training strategy is loading the pre-trained weights and then fine-tuning. The cost would be much lower than training a model from scratch. Are there any studies that examine ME under different training strategies?

2. ‘model extraction definitively works in that it is possible to approximate a blackbox model from just queries’. It would be better to add some conditions to this statement. It works on conventional deep-learning models (ResNet) on small-scale datasets. Nowadays, the model provided as APIs are normally foundation models or LLMs, would ME still work on those models?

3. It's unclear if the findings generalize to other types of ML tasks. Is all ME studies focus on classification tasks? It would be more insightful if this paper can expand the evaluation to include diverse models and tasks since this work is about thorough evaluations on ME attacks.

4. For vision tasks, the experiments are conducted using only ResNet models, like ResNet34, ResNet50, why not expand the diversity of the model architecture? More advanced architectures like large language models or foundation models are hoped to be examined.

5. The limitation of this work is hoped to be discussed.

---

> ### Author Response · Authors · 2024-09-27
>
> We thank the reviewer for the detailed comments.
>
> 1. We thank the reviewer for highlighting this point - perhaps the term “from scratch” was misleading in our work, and should be replaced with the term “from outsourced labeled samples”. Following the common methodology in the ME literature, we evaluate ME attacks by comparing the performance of a model trained using the original training dataset to that of a model trained on the attacker’s dataset. These two datasets might differ in two means: (i) data distribution - the original dataset is assumed to be sampled from the relevant task distribution, while the attacker’s dataset can be sampled from other, possibly distant, distributions; (ii) origin of labels - the original dataset is assumed to be labeled by some third-party (e.g., human annotators) and is considered to be well-annotated, while the attacker’s labels are in most cases obtained by querying the victim model. As such, when using the term “from scratch” we intended to describe the setting of training from the higher-quality well-annotated dataset, rather than the setting of training from randomly initialized weights. In our evaluations we considered both cases, where in the vision experiments, both the victim and the attacker model were trained from random weights, while in the language experiments, both models were pre-trained.
> The setting where the attacker can leverage some pre-trained model that was not available to the victim model owner represents another layer of reliance on prior knowledge, which we unified under the notion of having access to different amounts of training data samples. An attacker that can reduce the number of required queries by using pre-trained weights has the same dependency on prior knowledge as an attacker that does not have access to such weights but has more available samples.
>
> 2. Our statement described a general feasibility result that shows that ME attacks are possible under the query-access setting. As LLMs are essentially large-scale classification models we do believe that the same conclusions we draw for smaller-scale classification models will hold for such larger scale models as well, while the exact performance tradeoffs might naturally differ. This is also supported by other related works [1] showing that imitating another LLM requires substantially large amounts of queries.
> Specifically, as the majority of the ME literature focuses on smaller-scale models, it was natural for us to focus our discussion on these models as well, and shed light over a setting which has been extensively studied.
>
> 3. The majority of the ME literature indeed focused on classification models, and as mentioned earlier we naturally focused on the same settings as well. However there is evidence that similar phenomena can be observed on other tasks [1].
>
> 4. We focused our evaluations on victim models provided by known ME works. While it is interesting to evaluate more architectures, we believe we will obtain the same conclusions with slight variations in performance between settings.  It is important to note that given the scale of the study, computational overheads even for our current set of experiments are extremely large and took GPU months to run.
>
> 5. We will add a section to the paper on the limitations of the analysis. Namely, the study is empiric and only considers the settings that are commonly assumed in the academic literature. The findings are only applicable in settings where OOD detectors can be deployed, only for the models and training regimes assumed by the associated models. It is clear that there are settings where results will look differently, for example, in cryptanalytic extraction of fully connected networks [2]. Nonetheless, our study puts into question commonly used benchmarks for model extraction and suggests that risk assessment of model extraction may not be applicable to the use cases where data access is limited.
>
> In general, our extensive evaluations spanned across different settings commonly discussed in the ME literature, and we believe our conclusions are general enough to hold for other tasks, architectures, and scales. We hope this clarifies the reviewer’s concerns.
>
> [1] The False Promise of Imitating Proprietary LLMs. Gudibande et al. ICLR 2024.
>
> [2] Cryptanalytic Extraction of Neural Network Models. Carlini et al. CRYPTO 2020.

---

### Review · Reviewer_ph2c · 2024-08-21

**Summary Of Contributions:**

This paper studies whether black-box model extraction attacks pose a real threat. The authors study the effectiveness of attacks with bounded budget of in-distribution (IND) queries. They argue that reducing informativeness of out-of-distribution (OOD) means that the attacker needs to collect more IND data in sufficient quantity. In that setting, model extraction doesn't remain very cost-effective.

**Audience:**

Yes

**Broader Impact Concerns:**

Model extraction is a security concern and the authors should add an impact statement outlining the threat posed by model extraction and situate their contribution within it.

**Claims And Evidence:**

No

**Requested Changes:**

Please address the comments above.

The cost calculation of CIFAR10 in Sec 3.2 is not clear. Where did the numbers 35$, 25$, 50, etc. come from?

The authors have considered vanilla attacks that select samples uniformly from some distributions. There are sample-efficient techniques based on active learning even for model extraction as cited in the paper ("Exploring Connections Between Active Learning
and Model Extraction", "Activethief: Model extraction using active learning and unannotated public data"). These advanced techniques come at very little cost to the attacker. These need to be evaluated to see if the paper's claims hold under these attacks.

**Strengths And Weaknesses:**

I am not convinced by the conclusions drawn by the authors ("not a practical threat"). Though the paper is positioned as whether model-extraction is a practical threat, the assumptions made by the authors are more theoretical in nature than practical.

The authors assume that OOD data is uninformative. In practice, it is not too difficult for an attacker to select a distribution closer to the victim model's distribution. We do see in Sec 5.1 that attack with CIFAR100 is highly successful on a CIFAR10 model.

The authors take the number of queries to measure the cost of an attack. In domains like medical imaging, the cost of expert labeling could be orders of magnitude higher than the cost of querying a pre-trained model. Thus, an attacker could still benefit from firing a larger number of queries on a model than getting a medical expert to annotate a smaller number of queries.

The RQs do not convincingly show impractically of model extraction attacks.

---

> ### Author Response · Authors · 2024-09-27
>
> We thank the reviewer for the extensive review.
>
> ```"The authors assume that OOD data is uninformative. In practice, it is not too difficult for an attacker to select a distribution closer to the victim model's distribution. We do see in Sec 5.1 that attack with CIFAR100 is highly successful on a CIFAR10 model."```
>
> We do not assume OOD data is uninformative, but rather that it can be easily made uninformative using mechanisms such as the one we describe in section 4.3. As the attacker distribution grows closer to that of the victim distribution (such as in the case of CIFAR-10 and CIFAR-100 which are nearly the same distribution) it will naturally be more informative and less likely to be ``filtered’’ out without negatively impacting the model’s utility. However this access can be thought of as a form of prior knowledge over the distribution, which aligns with our claim over the dependency on prior knowledge.
>
> We want to also mention that our work considers the standard academic model extraction setting and the same criticism would apply to all of the prior work. That is to say, if it “is not too difficult for an attack to select a distribution closer to the victim model’s”, then the large number of papers that propose model extraction attacks based on unrelated datasets would have been irrelevant. Our work more broadly tries to figure out what makes model extraction attacks work.
>
> ```"The authors take the number of queries to measure the cost of an attack. In domains like medical imaging, the cost of expert labeling could be orders of magnitude higher than the cost of querying a pre-trained model. Thus, an attacker could still benefit from firing a larger number of queries on a model than getting a medical expert to annotate a smaller number of queries."```
>
> We agree with the reviewer’s comment. In section 5.5 we discuss this point and explain that we chose to answer the Q5 - “Can ME be used to reduce data labeling costs?” by the answer “sometimes” since it depends on the nature of the task and data modality itself. There are many use cases in which crowdsourcing annotations may be a cost effective alternative. There are also other cases, such as in the medical domain, where the data collection and labeling costs are most likely to be higher than that of querying the victim model.
>
> In the paper, we explicitly consider model extraction in the most widely used academic setting and argue that, just as the reviewer points out, sometimes these attacks are practical. Yet, as our work shows, they are not really practical in the currently used settings in academia.
>
> ```"The RQs do not convincingly show impractically of model extraction attacks."```
>
> In our work we addressed some of the claims and assumptions commonly used in the ME literature in order to provide a better understanding of the threat model of this class of attacks. Through our extensive analysis and evaluation we hope to emphasize the leading factors underlying the performance of ME attacks and highlight the need in refining the notion of ME attacks and their corresponding adversarial goals.
>
> We want to also highlight that we are not saying that model extraction attacks are impractical. We say that currently assumed threat models by the academic literature are focusing on a setting that is inherently biased and we suggest that the adversarial goals should be redefined.
>
> ```"The cost calculation of CIFAR10 in Sec 3.2 is not clear. Where did the numbers 35, 25, 50, etc. come from?"```
>
> These numbers are provided in the pricing information, at a time of writing, of Google Cloud and Amazon Sagemaker, both referenced in the text. Specifically, the cost of image labeling for classification tasks on the Google Cloud platform is 35 \\$ per 1k images for the first 50k queries (tier 1), and then reduced to 25\\$ per 1k images for the next 950k queries (tier 2).  Specifically, for the 60k images of CIFAR10 this would amount to 50k images being priced as tier 1 (50 * 35\\$) and the remaining 10k images priced as tier 2 (10 * 25\\$).

---

> ### Author Response · Authors · 2024-09-27
>
> ```"The authors have considered vanilla attacks that select samples uniformly from some distributions. There are sample-efficient techniques based on active learning even for model extraction as cited in the paper ("Exploring Connections Between Active Learning and Model Extraction", "Activethief: Model extraction using active learning and unannotated public data"). These advanced techniques come at very little cost to the attacker. These need to be evaluated to see if the paper's claims hold under these attacks."```
>
> We evaluated a set of attacks that represent main trends in ME literature. This includes DFME that generates data for querying yet to be explored regions, as well as KnockoffNets that uses existing dataset (ImageNet) but uses active learning methods for choosing the queries from within this dataset. ActiveThief operates in a similar manner to KnockoffNets, by also using ImageNet as the surrogate dataset and relying on active learning methods for sampling from it, and does not generate in-distribution samples. The sampling methods do differ, as ActiveThief does not assume knowledge of the ImageNet labels. As both methods use ImageNet samples, our evaluation, which allowed an even higher query budget, shows that using these samples still rely on OOD region informativeness (as discussed in Appendix G, with relation to figure 15).
>
> ```"Model extraction is a security concern and the authors should add an impact statement outlining the threat posed by model extraction and situate their contribution within it."```
>
> Model extraction attacks pose a significant threat to the security and privacy of machine learning models. By exploiting the model deployment, malicious actors can steal intellectual property (i.e. the model weights), undermine sensitive data privacy (e.g. launch stronger privacy attacks), and compromise the integrity of AI-powered systems (e.g. craft adversarial examples). In this work we challenged some of the common assumptions and motivations behind this class of attacks and discussed the conditions required for them to hold in practice, and showed that they often do not hold. We concluded by stating that the adversarial goals and incentives of this class of attacks should be redefined to make sure that benchmarks we use are appropriate to estimate the adversarial risk.

---

### Review · Reviewer_UHrb · 2024-10-06

**Summary Of Contributions:**

The paper challenges the common assumption in model extraction (ME) attacks that an attacker can save on both data acquisition and labeling costs by stealing a trained model through query access. They argue that current ME attacks rely on the adversary having prior knowledge of the victim model's data distribution, which dominates other factors such as the attack policy.
Through extensive experimentation, they demonstrate that prior knowledge of the attacker, specifically access to in-distribution (IND) data, is the primary factor influencing the success of ME attacks. The use of out-of-distribution (OOD) data has a minor impact on ME performance when mixed with IND data. When only IND data is used, the victim model serves primarily as a labeling oracle, rather than providing significant information about decision boundaries.

**Audience:**

Yes

**Claims And Evidence:**

Yes

**Requested Changes:**

- Extending the image classification findings to more datasets/architectures.

**Strengths And Weaknesses:**

## Strengths

- The paper focuses on an important problem of model extraction which can be useful in several practical use cases.
- The paper shows that the practical threat posed by ME attacks is often exaggerated, as the attacker bears comparable costs to the victim.
- Experiment results verify the claim that prior knowledge of the attacker, specifically access to in-distribution (IND) data, is the primary factor influencing the success of ME attacks.
- The paper is well written and easy to follow.

## Weaknesses
- Could the authors compare with some of the findings from Karmakar & Basu, 2023, where they build query-efficient model extraction attacks with public data and achieve high success rates?
- The model extraction attacks are limited to just one model arch and one dataset for image classification tasks. It would be interesting if the findings translate to other datasets/archs.

---

> ### Author Response · Authors · 2024-10-11
>
> We thank the reviewer for the detailed comments.
>
>  - Thank you for pointing Karmakar & Basu (2023) to us. The paper proposes a query sampling methodology that selects queries that maximize the disagreement between the target and the attacker model as well as maximize the entropy of the predictions of the attacker model. The attacker doesn’t not have access to the true data distribution but can sample from another natural distribution, such as Imagenet or AGNews. Although we did not evaluate this specific attack, we did evaluate other attacks with similar motivations. Namely, DFME that generates synthetic queries that maximize disagreement between the models, as well as, KnockoffNets that selects ImageNet queries based on a learned policy that optimizes for model disagreement, victim model confidence and label diversity. Our results show that these attacks heavily rely on the informativeness of the victim model’s responses to OOD queries, and when we limit this informativeness the attack success rate significantly drops.
>
>
>     We hypothesize that this will also hold for the attack proposed by Karmakar & Basu, as the attacker can not tell which queries get an informative response it will have to spend a significant portion of the query budget on learning uninformative regions. As we complicate, or even randomize, the predictions of the OOD region, the disagreement between the models would be almost always high in these regions and the attacker might not spend sufficient time in learning the actual IND region. Moreover, we mention that although the public datasets are not identical to the actual private dataset used by the victim model, they are not necessarily distributionally very different. For example, BBCNews and AGNews, just as CIFAR10 and CIFAR100, are relatively similar datasets, which can be considered a form of prior knowledge over the true distribution.
>
>  - For the image classification task we investigated 4 victim datasets - CIFAR10 in the main experiments and Indoor67, CUBS200, and Caltech256 when evaluating the KnockoffNets attack. We focused our evaluations on victim models provided by known ME works. While it is interesting to evaluate more architectures, we believe we will obtain the same conclusions with slight variations in performance between settings. It is important to note that given the scale of the study, computational overheads even for our current set of experiments are extremely large and take GPU months to run.

---

### Decision · Action_Editor_vCWW · 2024-11-26

**Recommendation:** Accept with minor revision

**Comment:**

I will list below a couple of minor comments I would like the authors to address.

1. You state that "We investigate the case where the query budget is constant and is set to the size of the original training dataset, i.e. the maximal query complexity still considered to be cost-effective.". Do I understand correctly that you consider querying the victim model to be as expensive as labeling the samples for all the other attacks except DFME? If so, I think it would be good to explicitly mention it.

2. I wonder if the answer to Q4 should also be "sometimes". Currently the answer is combined with the "OOD only reveals limited information" qualifier, but the question itself is not. If OOD queries are informative then I would imagine you could perform ME successfully (e.g. looking at the low $\tau$ results in Fig. 4).

Typos:
- "a ME" >> "an ME"
- "in generality" >> "in general"

**Audience:**

The ME attacks are an interesting and important study in the field of trustworthy ML, and hence I believe the paper is interesting for the TMLR audience. This sentiment was also expressed by all the reviewers.

**Claims And Evidence:**

This paper studies the effectiveness and cost of model extraction (ME) attacks. Authors demonstrate empirically that in order to obtain cost effective ME attacks, the attacker must have access to good quality data for the ME queries.

Authors study five research questions related to the cost/efficiency trade-offs in ME attacks, and provide empirical evidence on their claims using computer vision and NLP tasks. On computer vision evaluation, the results are presented for four data sets and for two ME attacks including the strong DFME attack. The results on all of the combination support authors claims. The evaluation for NLP is more limited, with evidence only for a single task. However, the findings on this task are in line with the ones observed in computer vision.

I believe the empirical evaluation, especially for the computer vision, is strong enough to support the claims. Also all the reviewers were satisfied with the evidence for the claims.

---

> ### Author Response · Authors · 2024-12-19
>
> Thank you very much for the constructive feedback.
>
> We changed Q4 to sometimes per your suggestion and clarified in the text, and fixed typos.
>
> Regarding the first comment, we are not sure we understood and would appreciate further clarification. We discuss the differences between the cost of labeling samples using crowdsourcing services and by querying the model in section 3.2 and in the answer to Q5, and in general conclude that ME attacks essentially serves as labeling oracles and as such are only cost effective in settings where labeling the data is cheaper by querying the model than by using crowdsourcing, which is not the case for many settings. We fix the query budget to be the size of the training set since in the setting where it is much larger than the training set, it is even more crucial that labeling using the model will be cheaper than other labeling solutions. We do not differentiate between the different attacks, but since DFME requires a large query budget to work we had to allow it to use a larger query budget to be even included in this comparison.

---

> > ### Comment · Action_Editor_vCWW · 2024-12-20
> >
> > Thanks for your response!
> >
> > My question was mainly about the " i.e. the maximal query complexity still considered to be cost-effective." part, and I was mainly wondering where does this "maximality" come from?